# Hand position fields of neurons in the premotor cortex of macaques during natural reaching

Sheng-Hao Cao[1,2,3], Xin-Yong Han[1,3], Zhi-Ping Zhao[4], Jian-Wen Gu[5], Tian-Zi Jiang [1,2,6] & Shan Yu [1,2,3,7] ✉

While hippocampus represents spatial information through place cells for body navigation, whether motor areas employ a similar framework to guide hand reaching remains unknown. Here, we investigate tuning properties in dorsal premotor cortex (PMd) during naturalistic reach-and-grasp tasks in four monkeys. We find that 22% (132/601) of PMd neurons increase firing rates when the monkey's hand occupies specific positions in space, forming the position fields. These cells represent the hand position highly efficiently, achieving ~80% accuracy for decoding hand trajectories with only 50 most dedicated position tuned cells ( ~ 10% of all recorded neurons). The hand position is co-represented with hand moving direction, speed, and reward location in the same population of PMd neurons, forming a mixed-selective framework to integrate positional and kinematic information. Our findings suggest field-like positional coding may be a mechanism shared across brain regions for spatial representation in goal-directed movements, including body navigation and forelimb reaching.

An essential function of the brain is to enable the planning and execution of goal-directed movements that can avoid obstacles and approach targets optimally. These movements can occur at various scales, such as the spatial navigation of the whole body and the execution of reaching using the hand. The ability to represent the spatial information of the body[1–6] or body parts[7–10] and to intergrade it with kinematic parameters is essential for guiding such movements.

For whole-body navigation, this ability is orchestrated by neurons in the hippocampus and its surrounding areas, exhibiting mixed selectivity among various task-relevant parameters, including position[5], head direction[11], moving speed[12], and goal orientation[13]. Specifically, position of the body can be represented by 1) place cells in the hippocampus, which discharge when the animal occupies specific

locations, giving rise to the concept of place fields[5], and 2) grid cells in the medial entorhinal cortex, characterized by periodic hexagonally spaced place fields[1]. Together, these place fields provide an efficient framework for encoding spatial information, aiding animals in constructing cognitive maps for navigation. For body part movements such as reaching with the hand, it has been known that parameters such as hand moving direction, speed, and goal position are represented through the mixed selectivity of neurons in motor relevant areas[14–16]. Also, neural activities in these areas can be modulated by the spatial location of a reach, either by the starting position[17], the ending position[18,19], or a combination of both[20]. However, it remains unclear whether a field-like representation, either place-cell like or grid-cell like, for hand positions exists somewhere within the motor areas. Resolving this question would offer insights into whether a general

[1]Laboratory of Brain Atlas and Brain-inspired Intelligence, Institute of Automation, Chinese Academy of Sciences, Beijing, China. [2]School of Artificial Intelligence, University of Chinese Academy of Sciences, Beijing, China. [3]State Key Laboratory of Brain Cognition and Brain-inspired Intelligence Technology, Chinese Academy of Science, Beijing, China. [4]Department of Ophthalmology, First Hospital of Jilin University, Changchun, China. [5]Department of Neurosurgery, The 9th Medical Center of Chinese PLA General Hospital, Beijing, China. [6]Xiaoxiang Institute for Brain Health and Yongzhou Central Hospital, Yongzhou, China. [7]School of Future Technology, University of Chinese Academy of Sciences, Beijing, China. ✉e-mail: shan.yu@ia.ac.cn

representation framework is employed by the brain to guide goal-directed movements across different scales.

To coordinate goal-directed hand movements, a network of various cortical areas is necessary, with the dorsal premotor cortex (PMd) being a crucial node. PMd neurons become active during the delay period before a movement[8,9], and disruption of PMd activity leads to impaired reaching performance[21,22]. PMd neurons exhibit directional tuning for hand movement[23–26] similar to that in the primary motor cortex, as well as tuning for other kinematic parameters, including speed[24,27], distance[28], and reaction time for a reach[29,30]. Moreover, PMd neurons are influenced by the relative position of the hand, eye, and target, enabling reference frames transformation during planning[10,31]. Given its role in movement planning and the integration of spatial and kinematic information, PMd presents an appealing target for exploring field-like representation for hand position.

In this study, we simultaneously recorded hand positions and spiking activities in the PMd of four rhesus monkeys (*macaca mulatta*) while they engaged in a naturalistic reach-and-grasp task. We analyzed neuronal activities from the perspective of spatial firing rate maps. We found that a small fraction of the PMd neurons (22%, 132/601) exhibited field tunings to hand position. We classified 8% of the PMd neurons (50 out of 601) as primary hand position-tuned cells based on spatial mutual information criterion and two distinct tuning reconstruction analyses. We found that the spatial tuning curves of these cells can be well fitted by a 2D Gaussian function instead of a linear plane. Furthermore, this hand position selectivity could not be fully explained by other known tuning properties in the PMd, such as hand moving direction, speed, or reward location, indicating that hand position information is a non-negligible component of the mixed selectivity in the PMd. Notably, such relatively modest neuronal sub-population achieves 80% of the decoding accuracy regarding hand moving trajectories. This finding underscores the pivotal role of the field-like hand position tuning in PMd during goal-oriented hand movements.

## Results

### Behavioral task and neuronal recording
We performed this study in four male macaques (*macaca mulatta*, Monkey A, B, X, and Z), each of which was implanted with a Utah Array (Blackrock Neurotech) in the left PMd (Fig. 1A). The array consisted of 96 electrodes for monkey X and 48 electrodes for other monkeys. Neurophysiological activities were recorded while the monkeys performed a naturalistic reach-and-grasp task[32,33] (Fig. 1B), during which they used their right arm to grasp fruit pieces on a wand held by the experimenter. In each trial, the wand was moved to an arbitrarily chosen location, either remaining static or changed to a new position while the monkeys were reaching (see Supplementary Fig. 1 for trajectories of the food). The position of the right hand of the monkeys was reconstructed from video recordings captured by four cameras (Supplementary Fig. 2A). In total we collected 839 (Monkey A, 51; Monkey B, 101; Monkey X, 390; and Monkey Z, 297) putative single units (SUs) during 18 daily behavioral sessions, each session lasting an average of $26.3 \pm 6.5$ min (mean ± SD) and comprising approximately 100 food trials. See Supplementary Table 1 for detailed information about each session.

### Identification of hand position tuning in the PMd
Figure 1C shows trajectories of the hand movements (black lines, projected to the XY plane, i.e., horizontal plane, most reaching movements were performed along this plane) for an exemplary session from each monkey, superimposed with the spikes of a representative cell. The observed spikes were not random; rather, neurons tended to discharge when the hands were within specific regions. To account for the effect of different time-occupancy in individual positions, we generated 2D-spatial firing rate maps by dividing the spike-

count map by the time-occupancy map (see Methods and Supplementary Fig. 3).

To quantify the spatial specificity of the firings of PMd neurons, we calculated the spatial information (SI)[34], defined as information content of individual spikes regarding the position of the hand, for each neuron (see Methods). A hand position-tuned cell was identified if its SI exceeded the 99th percentile threshold of the shuffled data. Figure 1C shows the 2D-spatial firing rate map of four exemplary cells (one for each monkey) that met such criterion. To ensure the within-session stability of the hand position tuning, we first selected stably recorded neurons by computing the Pearson correlation between spatial firing rate maps from the first and second halves of the session (see Methods and Supplementary Fig. 4). This process resulted in 601 stable cells (Monkey A, 40; Monkey B, 83; Monkey X, 248; and Monkey Z, 230). Among them, 132 cells (22.0%; Monkey A, 4; Monkey B, 34; Monkey X, 57; and Monkey Z, 37) were identified as the hand position-tuned cells. This proportion was significantly higher than expected by chance ($P = 0$, Binomial test with expected chance level $P_O = 0.01$), and the average SI was $0.59 \pm 0.29$ bits/spike (Fig. 1D). These results demonstrate that a sizeable proportion of PMd neurons exhibit significant hand-position specificity in their firing activities.

### Spatial tuning curve analysis
To examine the field-based tuning of hand position quantitatively, we first identified the hotspots within the spatial firing rate maps (above 50% of the peak firing rates) as the position fields, similar to the place fields in hippocampus cells[35]. Then we fitted the firing rate $r$ within the position field of each cell using a 2D Gaussian function of hand position: $r(x,y) \sim \mathcal{N}(\mu, \Sigma)$ (Fig. 1E, see Methods). As a result, the firing activities of 83% (109 of 132) of the hand position-tuned cells could be well described by a Gaussian function (coefficient of determination, $r^2 = 0.80 \pm 0.19$, significance level threshold 0.001, one-sided F-test). For comparison, we also fitted the firing activities as a linear function of hand position to explore whether these cells encoded spatial gradients at certain orientations, as previously suggested for neurons in the primary motor cortex[36]. Only 50% (66 of 132) of the hand position-tuned cells possessed a planar neuronal response surface within the 2D space ($r^2 = 0.52 \pm 0.24$, significance level threshold 0.001, one-sided F-test). More importantly, the coefficient of determination of the linear function was significantly lower than that of the Gaussian function ($P = 2.46 \times 10^{-26}$, $n = 132$ cells, one-sided paired t-test, Fig. 1F). The superiority of Gaussian fitting remained in the case of fitting full map data ($P = 3.02 \times 10^{-26}$, $n = 132$ cells, one-sided paired t-test, Supplementary Fig. 5A). These findings suggest that hand position is indeed represented by a field-like tuning of cells in the PMd. Additionally, we examined whether these position fields exhibited periodic hexagonal spacing and found few grid-like cells (see Supplementary Information).

### Mixed selectivity in the PMd
During the hand reaching movement, in addition to the hand position, other task variables could also affect neural activity, complicating the analysis of spatial firing rate maps. To address this issue, we first calculated the mean vector length, speed modulation depth, and food location SI to investigate the selectivity of PMd neurons for hand moving direction[23–26], speed[24,27], and food location[37,38], respectively (see Methods for identification criteria and Fig. 2A for exemplary cells for each cell type). The population summary of various tuning properties is shown in Fig. 2B. We found that PMd neurons exhibited mixed selectivity for diverse task-relevant variables. We identified 24% of PMd neurons (144 of 601) as hand direction-tuned cells and 32% of PMd neurons (192 of 601) as hand speed-tuned cells. We also found 3% of PMd neurons (18 of 601) showed spatial specific firings for food location. Notably, 36% of hand position-tuned cells (47 of 132) showed no significant correlation with other variables.

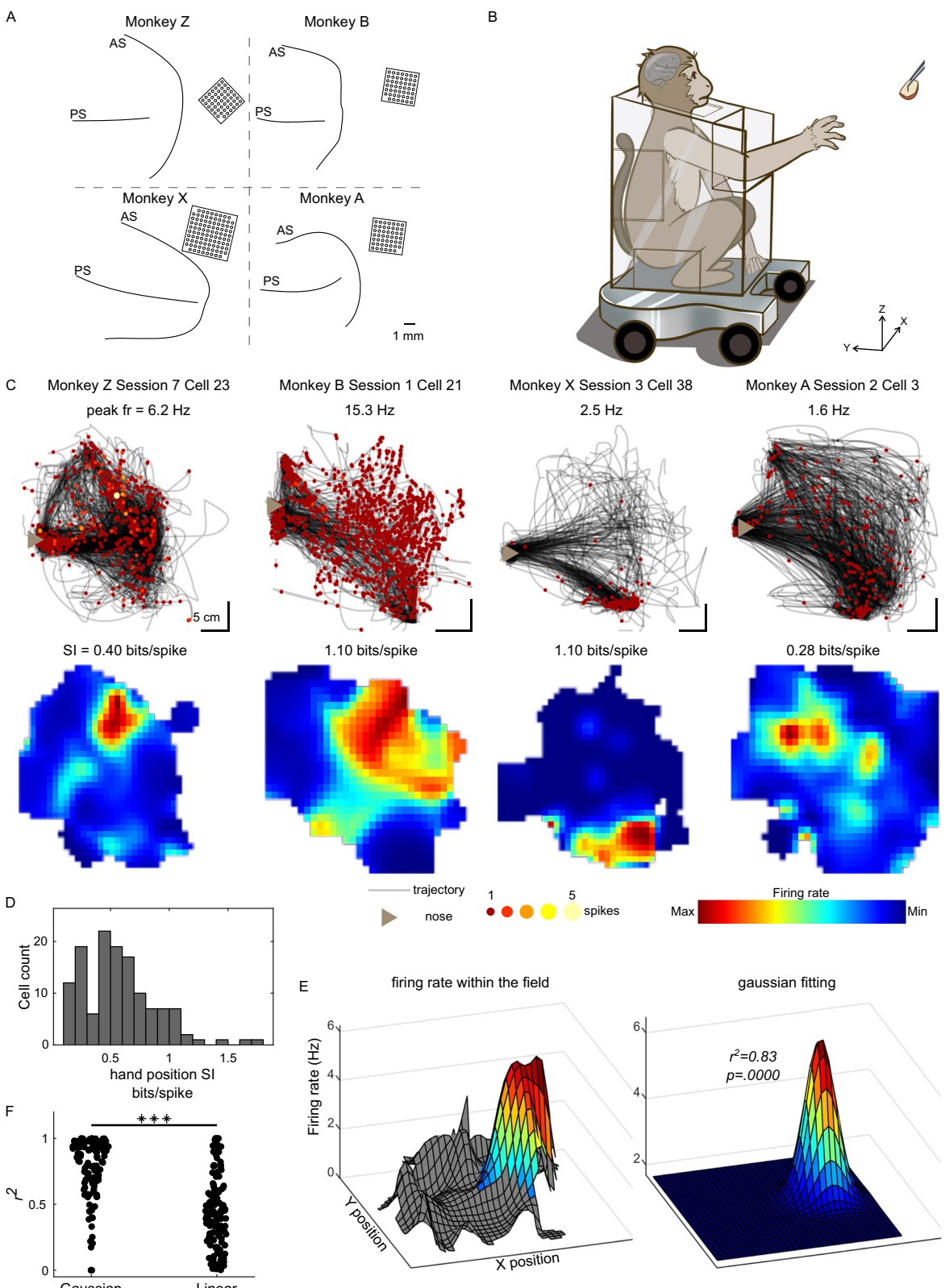

D

E firing rate within the field gaussian fitting

F

Next, to ensure that the spatial specific firing pattern we observed could not be explained solely by the tunings for the direction, speed or food location, we screened hand position-tuned cells through reconstruction analysis[13] (see Methods and Supplementary Fig. 6). We conducted a two-step analysis to successively assess the effects of kinematics and food location tunings. First, we retained 80 out of 132 candidate hand position-tuned cells which had lower reconstruction

errors assuming pure position tuning (hand-position/speed-direction index > 1, cf. the x-axis in Fig. 2C), which suggested that their position tuning property cannot be attributed to the observed speed-direction tuning. Second, out of the remaining 80 cells, we identified 50 (Monkey A, 0; Monkey B, 18; Monkey X, 17; and Monkey Z, 15) cells which had stronger hand position tunings than food location tunings (hand-position/food-location index > 1, cf. the y-axis in Fig. 2C), which are

**Fig. 1 | Hand position specific tuning in the dorsal premotor cortex. A** Recording sites mainly covered the hand area of the dorsal premotor cortex in each monkey. Black squares denote the position of inserted Utah arrays (96 electrodes for Monkey X, 48 electrodes for Monkey A, B, and Z). Black lines denote brain sulci used to locate the implantation site. PS, principal sulcus; AS, accurate sulcus. **B** Diagram of the naturalistic reach-and-grasp task. **C** Upper row: trajectories (black lines) of monkeys' hand from four example sessions (one for each monkey). Spikes from four exemplary cells in each corresponding session are plotted as circular markers. The color and size of the circles are set according to the spike counts. The brown triangle indicates the position of the monkey's nose (facing right). Peak firing rate for each cell is labeled on the top of panels. The vertical scale bars represent 5 cm along the x axis and the horizontal scale bars represent 5 cm along the y axis in each panel. Bottom row: corresponding 2D spatial firing rate maps, denoted with SI. Dark blue indicates the minimal firing rate within the map, and dark red indicates the maximal firing rate within the map. **D** The distribution of SI of hand position-tuned cells ($n = 132$). **E** Fitting the hand position firing fields using a 2D Gaussian function. Left, color-coded firing activity of the first cell in (**C**), shown as a function of hand position. The area outside the field is masked with gray. Right, fitted Gaussian function of the field, coefficient of determination, $r^2 = 0.83$, one-sided F-test, F-statistic = 170.75, $P = 0.0000$. **F** Comparison between the $r^2$ of Gaussian fitting and that of linear fitting, $P = 2.46 \times 10^{-26}$, $n = 132$ cells, one-sided paired t-test. *** $P < 0.001$. Source data is provided as a Source Data file.

referred as primary hand position-tuned cells hereafter (see Supplementary Fig. 7 for spatial firing rate maps for primary hand position-tuned cells). It is noteworthy that we do not suggest exclusive tuning for hand position in these cells. Instead, the designation underscores the hand position tuning as a non-spurious feature within the mixed selectivity framework in the PMd.

After excluding cells predominantly affected by other variables, we reevaluated the fitting results for the spatial tuning function of firing activities. The firing rate of these primary hand position-tuned cells continued to be better described as a Gaussian function of hand position across the space (full map fitting: $r^2 = 0.58 \pm 0.26$, field area fitting: $r^2 = 0.76 \pm 0.24$) rather than a simple linear plane (full map fitting: $r^2 = 0.27 \pm 0.14$, field area fitting: $r^2 = 0.40 \pm 0.29$; full map fitting comparison: $P = 9.35 \times 10^{-12}$ and field area fitting comparison: $P = 2.49 \times 10^{-11}$; $n = 50$ cells, one-sided paired t-test, Supplementary Fig. 5B).

### Characteristics of primary hand position-tuned cells

We next assessed other properties of spatial firing rate maps of the primary hand position-tuned cells. The spatial coherence, which measures the smoothness of firing rate maps, was $0.39 \pm 0.16$ (Fig. 3A). The mean spatial sparsity, defined as the fraction of motion space in which a cell is active, was $0.52 \pm 0.17$ (Fig. 3B). To further highlight the characteristics of the position fields, we imposed requirements on the sampling range and sampling point count of the fields in maps, resulting in 39 valid hand position fields (see Methods). The mean area of position fields was $98.7 \pm 85.3 \text{ cm}^2$ (Fig. 3C). Multiple fields were observed in 12% (6 of 50) primary hand position-tuned cells (Fig. 3D). The shapes of fields were predominantly elongated, as indicated by the ratio of the fields' principal axis lengths significantly larger than 1 (Fig. 3E, $P = 1.71 \times 10^{-12}$, $n = 39$ valid fields, one-sided paired t-test). For each monkey, we gathered valid fields across sessions and drew them together (Fig. 3F), showing that these fields tended to be widely distributed across the reachable space.

To further examine how the population of primary hand position-tuned cells could represent the hand positions across the reachable space, we used these cells to decode the hand moving trajectories (cell number $n = 50$, averaged $4.2 \pm 2.9$ across individual sessions, see Methods for the decoding process and Fig. 3G for exemplary observed and decoded trajectories). Overall, the correlation coefficient (cc) between the observed and decoded trajectories across all sessions in all animals was $0.54 \pm 0.10$ (primary hand position-tuned cells in Fig. 3H). We also evaluated the decoding performances using the group of all putative SUs ($cc = 0.68 \pm 0.17$, cell number $n = 839$, averaged $59.2 \pm 31.9$ across individual sessions, all cells in Fig. 3H) or the group of cells with stable spatial firing rate maps ($cc = 0.68 \pm 0.16$, cell number $n = 601$, averaged $42.8 \pm 18.6$ across individual sessions, stable cells in Fig. 3H). Although lower than the performance using all cells ($P = 0.0046$, $n = 18$ sessions, one-sided paired t-test) or stable cells ($P = 0.0034$, $n = 18$ sessions, one-sided paired t-test), the primary hand position-tuned cells, which amount to only 10% of the recorded neurons, achieved 84% of the decoding performance obtained by using all cells and 82% of the decoding performance obtained by using stable

cells. However, we note that it does not mean that the remaining non hand position tuned-cells—which constitute 90% of the recorded neurons—only contain less than 20% of the hand positional information, as such information may be represented in a redundant way across the whole population. We randomly selected a subset of cells matched in quantity to the primary hand position-tuned cells after we removed hand position-tuned cells from all cells (random-select cells1 in Fig. 3H) or stable cells (random-select cells2 in Fig. 3H) to decode hand trajectories (random-select cells1: $cc = 0.31 \pm 0.18$, random-select cells2: $cc = 0.33 \pm 0.18$; see Supplementary Fig. 8 for exemplary decoded trajectories with random-select cells). Both random-select cell groups demonstrated significantly lower decoding performance compared to the primary hand position-tuned cells (Fig. 3H; random-select cells1: $P = 1.89 \times 10^{-4}$, random-select cells2: $P = 3.37 \times 10^{-4}$; $n = 18$ sessions, one-sided paired t-test). What the above results emphasize is that the primary hand position-tuned cells represent the spatial information during reaching highly efficiently.

## Discussion

Graziano et al. observed that long-period electrical microstimulation of various sites in the precentral cortex (including the premotor and the primary motor cortices) induced the movement of the monkeys' hands to specific spatial positions[39]. Our identifications of field-based tuning for both hand and target positions reveal the spatial information encoding framework that likely underlies the observed effects of microstimulation (see Supplementary Fig. 9 for the spatial organization of positional preferences in the recording sites). The field-based tuning of hand position we found in the premotor area appears different compared to that observed in the primary motor area and parietal cortex, which has been suggested to exhibit gradient tuning, i.e., cells encode not specific hand positions but rather position gradients along certain orientations[36,40,41]. We note that the reaching task employed in our study encompasses a considerably broader reachable space compared to that utilized in previous works and aligns with the natural conditions of monkey's arm movements better. In a confined space, the field-based tuning can appear as the linear gradient tuning along certain orientations. Consequently, future research efforts may be warranted to reassess the hand position tuning in the motor area under more natural and expansive conditions.

Several studies have shown that the spatial positions of a reach movement in the 3D space would modulate existing tuning properties in motor areas, such as direction tuning[20,26,42], distance tuning[17]. Supposing 3D spatial information is similarly encoded in neural activities in the form of position field, discretely sampling trajectories on the 3D spatial firing rate maps would provide a possible explanation for such observations.

As previous studies have revealed the important role of the PMd in coordinate transformation[10,31], the current finding of the hand position tuning in the PMd provides a straightforward way to realize such transformation computationally. By integrating the spatial information of target/stimuli in the PMd and related areas[43,44] and the hand position represented in this study, it is possible to obtain a vector representation of the direction and distance of a reaching

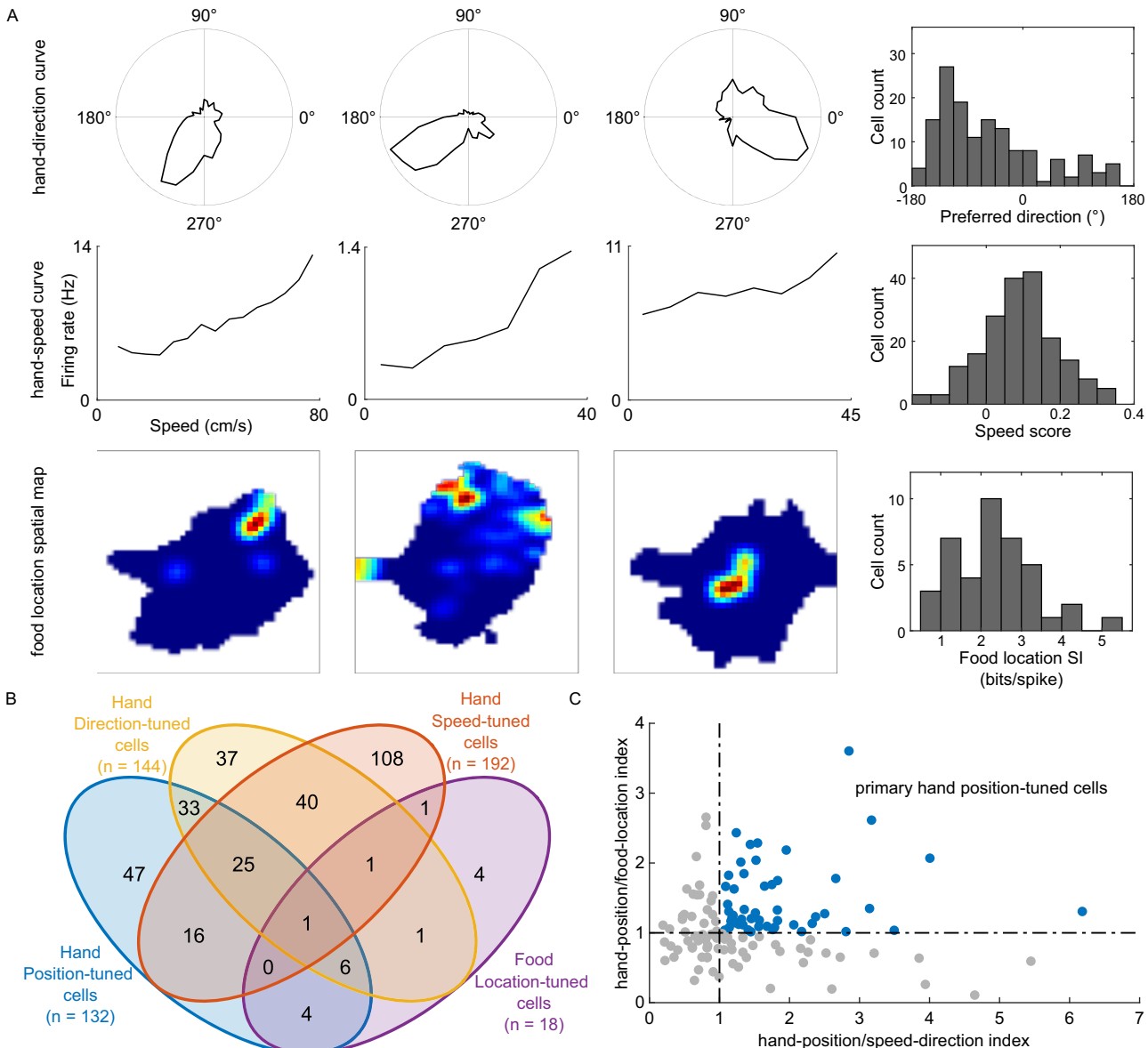

**Fig. 2 | Hand position specific firing in PMd cell is not a byproduct of known tuning properties. A** Firing rate as a function of hand-direction, hand-speed, and food-location in three exemplary cells, respectively. The upper row, firing rate as a function of hand moving direction, followed by the distribution of preferred direction of hand direction-tuned cells ($n = 144$). The middle row, firing rate as a function of hand moving speed, followed by the distribution of speed score of hand speed-tuned cells ($n = 192$). The bottom row, color-coded spatial rate maps for food location, followed by the distribution of SI of all food location-tuned cells ($n = 18$). Dark blue indicates the minimal firing rate within the map, and dark red indicates the maximal firing rate within the map. **B** Population summary of cells tuned to hand position, hand moving direction, speed, or food location. **C** Scatter plot showing hand-position/speed-direction indexes and hand-position/food-location indexes of the 132 hand position-tuned cells which met the SI criterion. During the reconstruction analysis, we obtained two normalized reconstruction errors based on two opposing hypotheses and subsequently, the ratio of these errors yields the hand-position index. We identified cells with higher hand-position index ( > 1) as primary hand position-tuned cells whose hand position tunings could not be explained by pure other tunings ($n = 50$, blue dots). The vertical dotted line and the horizontal dotted line indicate the identity hand-position index. Source data is provided as a Source Data file.

movement through making populational subtraction based on the same spatial coordinate. This subtraction operation achieves a transformation from body-centered (position coordinates of the hand and the target) to limb-centered (direction and distance relative to the hand) reference framework. Then downstream cortical areas could further translate such information into intrinsic motor commands. Critically, as monkeys can execute consistent reach-and-grasp movements across varying body locations and orientations, we hypothesize that the field-like coding of hand position in the PMd provides robust hand positional encoding in egocentric frame[45], thereby establishing a computational foundation for stable body-to-limb coordinate transformations.

The simplified, natural reaching tasks we employed lacked explicit and distinguishable periods demanding preparations or planning. This leaves the questions of how these hand position tuned-cells are involved into the broader functions because PMd plays a significant role in many motor cognitive processes including preparation and execution of movement[46–48], abstraction[49], decision signals[50,51], and motor learning[52], etc. We derived our main findings based on the non-stationary period data, while another decoding analysis showed that populational hand position coding remained stable and would not be affected by the hand motion state (Supplementary Fig. 10B), suggesting that the representation of current hand position by these cells was not dependent upon the presence of

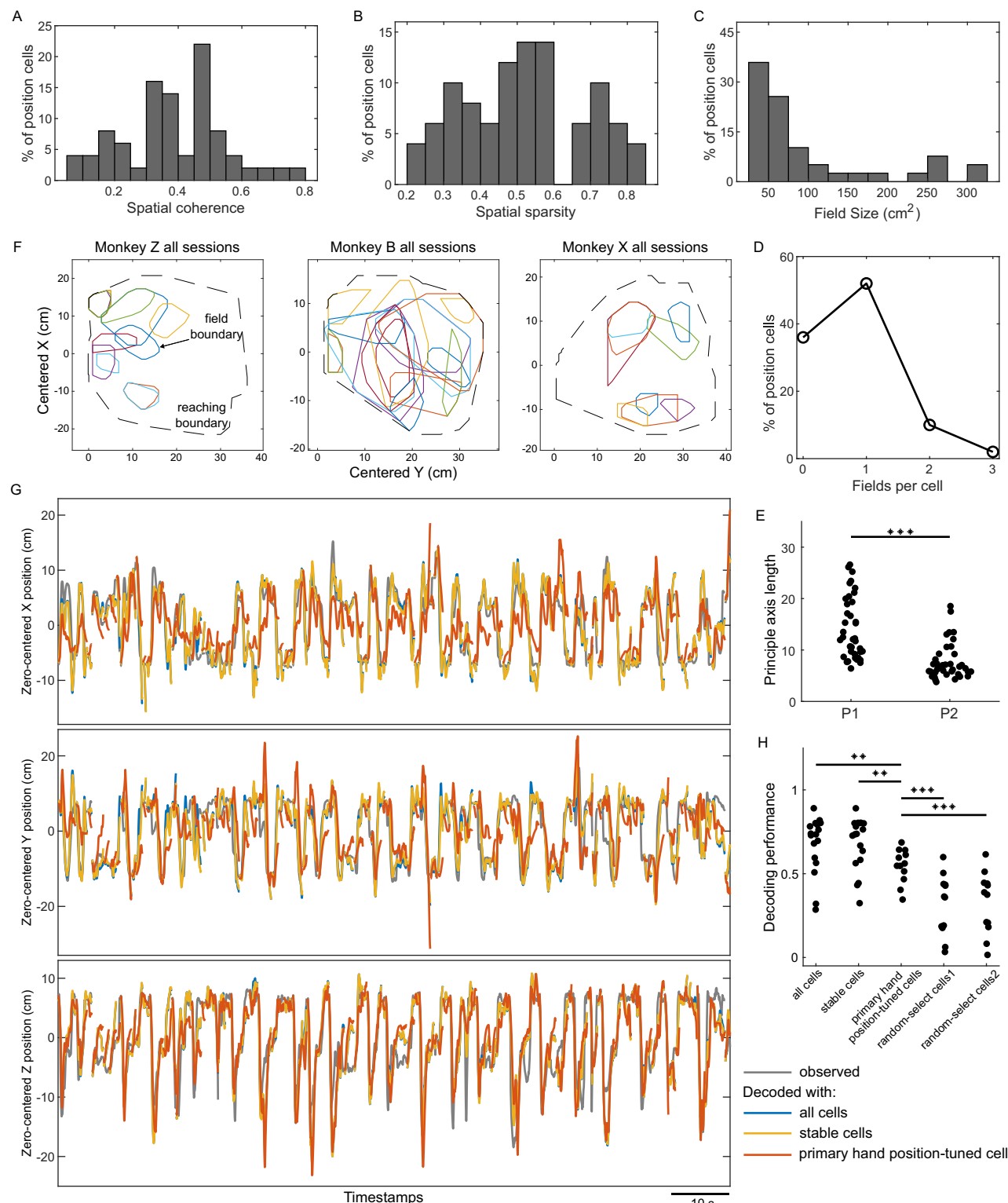

**Fig. 3 | Properties of primary hand position-tuned cells.** Distributions of spatial coherence (**A**), spatial sparsity (**B**), field size (**C**), and number of position fields per cell (**D**) across all primary hand position-tuned cells ($n = 50$). (**E**) Comparison of lengths of two principal axes of valid position fields ($P = 1.71 \times 10^{-12}$, $n = 39$ valid fields, one-sided paired $t$-test). **F** Distributions of the valid position fields across all sessions for each monkey. Colored lines distinguish fields from different hand position-tuned cells. These positions are relative to the nose position of the monkey, which was set as (x, y) = (0, 0). Dashed black lines indicate monkeys' reaching boundaries. **G** Examples of observed and decoded hand trajectories using different cell groups. Gray lines denote observed trajectories, blue lines denote decoded trajectories using all cells, yellow lines denote decoded

trajectories using stable cells, and red lines denote decoded trajectories using primary hand position-tuned cells. The results shown are based on the entire testing dataset for one session from Monkey B, with the stationary period excluded. The horizontal scale bar represents 10 s. Trajectories along the Z axis are calculated from the XZ plane. **H** Comparison of decoding performances using different groups of cells. All cells vs. primary hand position-tuned cells: $P = 0.0046$; stable cells vs. primary hand position-tuned cells: $P = 0.0034$; primary hand position-tuned cells vs. random-select cells1: $P = 1.89 \times 10^{-4}$; primary hand position-tuned cells vs. random-select cells2: $P = 3.37 \times 10^{-4}$. $n = 18$ sessions, one-sided paired $t$-test for comparisons. ** $P < 0.01$, *** $P < 0.001$. Source data is provided as a Source Data file.

motion itself. But it awaits future studies to reveal how the position coding framework we found here may contribute to the state estimation or motor planning, for example, exploring the dynamic properties[40,41,53,54] of hand position tuning, in which the coding is calculated for not only the instantaneous hand position but also the future and past positions.

In addition to the task-related variables explicitly analyzed in this study, eye movement is a potential confounding factor, as a few electrodes were located near the rostral part of the PMd, which might engage in oculomotor control[55]. However, we note that majority of electrodes with hand position-tuned cells were located away from the frontal eye fields (see Supplementary Fig. 9). In addition, we observed that the monkeys' gaze position during the experiments were primarily around the food reward rather than their hands, suggesting that the eye movements unlikely have played a significant role in the hand position tuning we identified. Nevertheless, incorporating a gaze fixation protocol or synchronized eye-tracking in future experiments will help clarify this issue.

Elucidating the specific tuning function of individual neurons can yield more insights into representational geometry and neurocomputing[56]. For example, it has been demonstrated that the nonlinear conjunctive selectivity of multiple features ensures improved reliability and efficiency in information representation compared to the pure selectivity[57]. In this context, the extrinsic space can be decomposed into two separate dimensions (or three dimensions in the 3D space). As a result, the theory of mixed-selectivity suggests that field-based tuning of position, which is a nonlinear mixed representation of various space dimensions, provides computational advantages over position gradient tuning, which entails pure selectivity along a single gradient dimension. The functional benefits of field-based tuning may account for the observed similarities in the tuning for 1) hand position during limb movements, 2) body position during navigation, and 3) the relative position of various concepts in an abstract cognitive map[58,59]. The advantages of field-based tuning in representing information are consistent with the high efficiency demonstrated in decoding hand trajectories in the present study, as well as the high efficiency in representing spatial or abstract conceptual information within a more general framework termed the Tolman-Eichenbaum Machine[60], suggesting that functional optimization may lead to a similar framework for information representation in different brain regions. In the same light, the mixed selectivity among position and kinematic parameters can be regarded as a more sophisticated conjunctive representation, constituting a higher-level representation framework shared between the hippocampus and motor-related areas to guide goal-directed movements.

While our findings emphasize the representational nature of hand position coding in PMd, an alternative interpretation emerges from dynamical systems theories of motor control. Recent studies probing the neural dynamics of PMd and related motor areas[48,61–64] demonstrate that population activity during reaching reflects low-dimensional, structured manifolds, rather than explicit encoding of kinematic variables. This raises the possibility that the observed hand position tuning may be a consequence of the need for a dynamical system to possess representations. Notably, such dynamics-centric frameworks do not negate the functional relevance of hand position coding. Instead, the mixed selectivity combining hand position tuning with other kinematic tunings may provide a foundation for implementing low-dimensional neural computations[15]. These representations and their geometric structures may serve as initial conditions[65,66,67,68] or boundary constraints that shape the morphology of neural manifolds[56].

## Methods

The experimental plan was approved by the Biomedical Research Ethics Review Committee of the Institute of Automation, Chinese Academy of Sciences (approval number: IA-201904). All experimental and surgical procedures involving animals were performed in accordance with the NIH Guide for the Care and Use of Laboratory Animals.

### Subjects and behavioral task setup

Four male rhesus macaques (Macaca mulatta), aged from 9 to 11 years were used in this study. Before the start of the experiments, all animals were surgically implanted with a titanium head post. The implantation surgery was performed under sterile conditions with isoflurane anesthesia. Animals were fasted 24 h before the surgery. Ketamine (10 mg/kg) was given intramuscularly for initial sedation and anesthesia. Then, animals were transferred to a dedicated surgery room and fixed in a stereotaxic apparatus. Gas anesthesia was performed with oxygen (30% partial pressure), nitrous oxide (70% partial pressure), and 1–2.5% isoflurane through endotracheal intubation. Adequate anesthesia was monitored and maintained throughout the procedure, and a titanium head post was secured to the skull with titanium cortical screws. Postoperative routine antibiotic therapy continued for 5-7 days. After surgery, animals were allowed to recover for 4 weeks to ensure stable fixation of the head post to the skull. Following veterinary approval of post-surgical recovery, monkeys were then trained to sit head-fixed in a primate chair for behavioral and electrophysiological experiments.

The behavioral task was a naturalistic reach-and-grasp task[32,33]. Specifically, fruit pieces were attached to one end of a wand manually held by the experimenter. The monkeys were allowed to use their right arm to reach and grasp the fruit pieces and then feed themselves. There was no explicit instruction or time limit on performing the task, which allowed the animals to choose how to reach and grasp at their own pace once the fruit pieces were presented. The wand was moved to random locations in front of the monkey, either staying static or changing to a new position during the reaching. The food might be moved outside the reachable range of monkeys to encourage their maximum reach movements. Importantly, these periods constituted only a very small proportion of each session as monkeys would not reach out if they found the food beyond their reaching limits for too long. There were no explicit constraints, except for those imposed by sitting in the primate chair with head fixed, on the spatial range of reaching for monkeys during the task. The recorded spatial ranges for each monkey were: Monkey A, 35 cm × 37 cm × 42 cm; Monkey B, 35 cm × 35 cm × 39 cm; Monkey X, 39 cm × 38 cm × 50 cm; and Monkey Z, 44 cm × 38 cm × 50 cm. In each daily session, about ~100 pieces of fruits were given, which lasted for 9.5 ~ 35.7 min. Each monkey was repeatedly tested for 3-8 sessions.

### Electrodes and implantation surgery

Intracortical microelectrode arrays (Utah Array, inter-electrode distance: 400 μm, electrode length: 1 mm, Blackrock Neurotech) targeting the hand area of the left dorsal premotor cortex (PMd) were implanted in the monkeys, with the same surgical procedure described above. For implantation of arrays, the principal sulcus and the arcuate sulcus were identified after the craniotomy. Then a multi-channel array (96 electrodes for Monkey X, 48 electrodes for Monkey A, B, and Z) was implanted in the PMd. Postoperative routine antibiotic therapy continued for 5-7 days. All animals underwent a 1–2 week recovery period, and their condition was approved by a veterinarian before behavioral test and recording sessions resumed.

### Hand motion tracking

Motions of the right hand of the animals were recorded using four cameras (Hikvision) from different angles (Supplementary Fig. 2A). The video data were sampled at 60-100 frames/sec. We firstly used markerless tracking software *DeepLabCut* (DLC)[69] to extract the 2D positions of the right hand (the metacarpophalangeal joint of the middle finger) and the food in each video frame and then reconstructed the 3D positions out of high confidence 2D data points

(likelihood ≥ 0.9, DLC estimated a likelihood value for each data extracted) using the toolbox *pose3d*[70]. In addition, we extracted the positions of monkeys' noses referred to as the position of the monkeys. Position data were then post-processed to fill small gaps (<150 ms) by median filtering. Camera calibration parameters for each camera pair were estimated after each recording session via the *Computer Vision Toolbox* from Matlab (Mathworks Inc.). We excluded the stationary periods during which the speed was lower than 1 cm/s to avoid the effects of changes in moving status[12].

## Neural recording and spike sorting

Neural data were recorded during the task. We used a CerePlex Direct system (sample rate: 30 kHz, Blackrock Neurotech) or an Intan Recording Controller (sample rate: 20 kHz, Intan Inc.) to save the raw, broad-band signals recorded from the implanted arrays. Units were extracted offline by the automatic spike-sorting software— Mountainsort[71]. A fifth-order Butterworth bandpass (250–7500 Hz) filter was applied to the raw signal, followed by a zero-phase component analysis (ZCA) whitening across channels. The detection threshold was set to −5.5 s.d., and the waveform length was 1.6 ms (−0.4 ms -1.2 ms). We further verified the sorted units manually in Offline Sorter (Plexon Inc.) and identified putative single units according to two criteria: a) 2-ms inter-spike interval (ISI) violation rate <2%; b) a clear clustering and separation of waveforms could be seen in at least one feature space. Only single units with an average firing rate > 0.1 Hz were included for further analyses.

Given that the recordings were obtained from chronically implanted electrode arrays, we also considered a case where unique cells were recorded repeatedly over multiple sessions and explored the effect on the hand position tuning. Please see Supplementary Information.

## Synchronization between videos and neural recordings

The camera (Hikvision) we used supports hardware-trigger mode, that is, when the camera receives a rising edge voltage (TTL 0 to TTL 1), it takes one frame image. We used a signal generator (DG1022U, RIGOL Technologies) to generate a periodic square wave signal to control the frame rate of the camera. We connected four cameras to the same output port of the signal generator, so we could record multiple videos from different angles simultaneously. Likewise, the trigger-in port of the neural recording system (CerePlex Direct or Intan Recording Controller) was also connected to the same output port of the signal generator, as a result, when a rising edge was generated, cameras captured one frame image while the neural recording system recorded the timestamp of this rising edge which also referred as the timestamp of this frame along with neural data.

## Calculating motion data and firing rate

After extracting the position data from the videos, we could further calculate the moving velocities and moving directions in each frame easily, so we got the position train, speed train, and direction train. We used the triggered timestamps in neural data to set the time bins to count the number of spikes resulting in the corresponding spike count train. We could divide the spike count train by the frame rate of videos to get the spike firing rate. We used the position trains and corresponding spike count trains to visualize the spike-trajectory plots and calculated the spatial firing-rate maps further. We used the speed trains and firing rate trains to calculate the speed-tuning curve. We used the direction trains and firing rate trains to calculate the mean vector length. We used the position trains and firing rate trains in our trajectory decoding analysis.

## Spatial firing rate map

Analyses were conducted in the XY (i.e., horizontal) plane as most reaching movements were performed along this plane.

Hand spatial firing rate map: spatial firing-rate maps were constructed for each putative single unit. Specifically, the spike counts and the time occupancy of the right hand in each spatial bin (1 cm × 1 cm) were calculated, generating one spike-count map and one time-occupancy map. Each map was then smoothed by a 2D Gaussian filter (standard deviation: 1.5 bin, Matlab function *imgaussfilt*). We then computed the firing-rate maps used for further analyses by dividing the spike-count map by the time-occupancy map. Invalid spatial bins with time occupancy less than 50 ms (less than 3 sampling points in the spatial bin) were excluded from the 2D firing-rate maps.

Food spatial firing rate map: we calculated the food firing rate map for each neuron as a function of the food location. We excluded data after the monkey grasped the fruit in each trial because the trajectories of the monkey's hand and food overlapped during this period. Then the computing process was the same as above.

## Recording stability

To test the stability of hand spatial firing rate maps, we divided all recorded data within each session into two equal halves. Two new firing-rate maps were computed, respectively with the same process mentioned above. The Pearson correlation, $r$, between the two maps was calculated. Cells with low stability, i.e., $r < 0.3$, were excluded from further analyses[72].

## Spatial firing activity fitting

Assuming that the firing rate $r$ is a 2D Gaussian function of the hand position $(x, y)$: $r(x, y) \sim \mathcal{N}(\mu, \Sigma)$. First, we rotate the positional coordinate axis by an angle of $\theta$ to eliminate the correlation between them:

$$\begin{bmatrix} u & v \end{bmatrix} = \begin{bmatrix} x & y \end{bmatrix} \begin{bmatrix} cos(\theta) & -sin(\theta) \\ sin(\theta) & cos(\theta) \end{bmatrix} \quad (1)$$

Then, we fitted the firing rate $r$ as a function of $(u, v)$ based on the equation below:

$$r(u, v) = a * exp - \left[ \left( \frac{u - \mu_u}{\sigma_u} \right)^2 + \left( \frac{v - \mu_v}{\sigma_v} \right)^2 \right] + b \quad (2)$$

where $a, b$ are gain and baseline respectively, $\mu_u / \mu_v$ is the mean value along each axis and $\sigma_u / \sigma_v$ is the standard deviation along each axis.

We extracted the rotation angle $\theta$ by using Matlab function *regionprops*. The fitted Gaussian function was obtained by MATLAB function *fit*. As a comparison, we directly used Matlab function *regress* to calculate the linear regression results.

## Valid position field detection

Firing rate maps were binarized by thresholding at 50% of the peak firing rate. Then we defined the valid position field as a set of contiguous regions whose area was larger than 25 cm² and was visited by the hand at least five times[35]. Field properties such as boundary, size, principal axis length, and eccentricity were obtained by using the MATLAB function *regionprops* applied to the binary maps.

## Spatial sparsity

Spatial sparsity was used to reflect the fraction of the motion space in which a cell is active[73], which was calculated as follows:

$$Sparsity = \frac{\left( \sum P_i \lambda_i \right)^2}{\sum P_i \lambda_i^2} \quad (3)$$

where $P_i$ denotes the occupancy probability in the $i$th bin, $\lambda_i$ denotes the firing rate of the cell in the $i$th bin.

## Spatial coherence

Spatial coherence was estimated as the mean correlation between the firing rate of each bin and the mean firing rate in the eight adjacent bins[74], which was used to measure the similarity of local firing rates to that of the neighboring bins.

## Measurements used for cell type identification

Spatial mutual information: the information theory method was used to quantify the amount of information transmitted by the firing of a neuron (bits/spike) regarding the interested experimental variables[2,34,75]. Specifically, we calculated the spatial mutual information as:

$$I = \sum_i P_i \frac{\lambda_i}{\lambda} \log_2 \frac{\lambda_i}{\lambda} \tag{4}$$

where $P_i$ denotes the occupancy probability in the $i$th bin, $\lambda_i$ denotes the firing rate of the cell in the $i$th bin and $\lambda$ denotes the overall mean firing rate of the cell.

SIs were calculated based on the hand spatial firing rate maps to select putative hand position-tuned cells and were also calculated based on the food spatial firing rate maps to identify the food location-tuned cells.

Speed modulation depth (to identify hand speed-tuned cells)[12]: we first calculated the speed-tuning curve according to the time series of moving velocity and the firing rate. The equally spaced bin size is 5 cm/s. The modulation depth of a cell was defined as the difference between the maximum and minimum firing rates in the speed-tuning curve. We also computed the speed score as the Pearson correlation between the time series of the velocity and the firing rate.

Mean vector length (to identify hand direction-tuned cells)[12]: first, the circular hand direction tuning curves (angle bin width π/16, Gaussian smoothing, standard deviation: 1.5 bin, Matlab function *smoothdata*) were calculated. Then, the mean vector length was computed as follows:

$$\left\| \frac{\sum r_k e^{i\theta_k}}{\sum r_k} \right\| \tag{5}$$

where $\theta_k$ is the angle expressed in radians in the $k$th bin and $r_k$ is the corresponding firing rate. ||*|| indicates the modulus.

## Shuffling and identification criteria

To obtain the shuffled data, spike trains were shifted by a random interval that was more than 30 s and less than the session duration minus 30 s (wrapping the end circularly to the beginning). Then we recalculated all measures mentioned above and repeated this procedure 500 times. If the raw measure was larger than the 99th percentile of the shuffled data, the cell was classified as the corresponding cell types and the chance level was set as $P_O = 0.01$. Measurement comparison was made within each session. It is worth noting that the nature of the speed modulation depth does not allow for the mixture of information coming from different cells, so that every individual cell had its own threshold.

## Reconstruction analysis

Inhomogeneous reaching movements could mislead our judgment of hand-position-tuned cells. For example, if a cell is only tuned for hand moving direction and speed, and the monkey's hand always passes through certain spatial locations with specific speed and/or direction, then an apparent tuning for spatial locations could be seen. To exclude such possibility, the reconstruction analysis[11,13] was adopted to examine how well the position tuning we observed can be explained by the kinetic parameter tuning properties. Specifically, this analysis was carried out by a two-step process: 1) assuming that the cell is only

jointly tuned to speed-direction and reconstructing the spatial firing rate map based on this assumption; 2) assuming that the cell is only tuned to hand location and reconstructing the speed-direction firing rate map.

The reconstructed spatial firing rate map was computed using the following equation:

$$\hat{r}(x,y) = \frac{\sum_{v,\theta} (p(v,\theta,|,x,y)*r(v,\theta))}{\sum_{v,\theta} p(v,\theta,|,x,y)} \tag{6}$$

where $p(v,\theta,|,x,y)$ is the fraction of time spent at a specific speed-direction bin $(v,\theta)$ while the hand in a particular spatial bin $(x,y)$, and $r(v,\theta)$ is the firing rate for this speed-direction bin.

The reconstructed speed-direction firing rate map was computed using the following equation:

$$\hat{r}(v,\theta) = \frac{\sum_{x,y} (p(x,y,|,v,\theta)*r(x,y))}{\sum_{x,y} p(x,y,|,v,\theta)} \tag{7}$$

where $p(x,y,|,v,\theta)$ is the fraction of time spent at a specific spatial bin $(x,y)$ while the hand in a particular speed-direction bin $(v,\theta)$, and $r(x,y)$ is the firing rate for this spatial bin.

The reconstruction error was defined as the normalized mean squared error (Eq. 8) between the observed tuning map and its reconstructed version. All maps were normalized by their maximal firing rate.

$$error = \frac{<(reconstructed - observed)^2>}{\max(observed) - \min(observed)} \tag{8}$$

We compared the errors of the two reconstructions mentioned above. In other words, we calculated the hand-position/speed-direction index = error under speed-direction-tuning assumption/error under hand-position-tuning assumption. The reconstruction with the lower error compared to its counterpart indicates that the property used to perform this reconstruction explains the activity tuning better than the other way around, for example hand-position index greater than 1 for stronger hand-position-tuned cells.

We adopted the same reconstruction analysis utilizing hand spatial firing rate maps and reward spatial firing rate maps to assess the interference from food location tunings.

Speed-direction firing rate map: we computed speed-direction firing rate maps to represent joint speed-direction tuning properties. The spike counts and time occupancy in each equally spaced bin (bin size: 5 cm/s for speed and π/8 for direction) were calculated. Each map was smoothed by a 2D Gaussian filter (standard deviation 1.5 bin). We then calculated the firing rate maps by dividing the smoothed spike-count map by the smoothed time-occupancy map. Bins in which time occupancy was less than 50 ms were excluded from further analyses.

## Trajectory decoding analysis

A linear Kalman filter was used to predict the trajectories of the monkey's hand during each session. Firing rates were calculated in partially overlapping 100 ms bins successively shifted by 50 ms (we downsampled the frame rate of motion data to 20 Hz for convenience). The *state* vector of the Kalman filter $X_t = [pos\ vel\ acc]^T$ describes the kinematic parameters of the hand in each time step $t$, and the *observation* vector of the Kalman filter $Z_t = [z_1 z_2 \ldots z_c]^T$ represents the firing rates of $c$ selected cells.

The generative model of the observation is formulated as:

$$Z_t = HX_t + Q \tag{9}$$

where $H \in \mathcal{R}^{c \times 6}$ is the observation matrix, and $Q \sim N(0, \mathcal{R}^{c \times c})$ is the observation noise. The state propagating process is formulated as:

$$X_{t+1} = AX_t + W \tag{10}$$

where $A \in \mathcal{R}^{6 \times 6}$ is the coefficient matrix and $W \sim N(0, \mathcal{R}^{6 \times 6})$ is the system noise. Parameters are estimated using the least square method[76].

We used three different groups of neurons to decode the trajectories in each session: group 1—all cells, all the putative single units after spike sorting; group 2—stable cells, cells with stable hand spatial firing rate maps; group 3—primary hand position-tuned cells, cells met the criteria. We utilized 70% of the data starting from the session beginning to train the model, and the remaining 30% of the data for testing. To decode the trajectories along the z-axis, we trained another decoder in the XZ plane. The incorporation of the x-axis data helped improve performance in the z-axis. The average correlation coefficients between the predicted and observed trajectories in test datasets were used to quantify the decoding performance.

We directly applied the trained decoders mentioned above to the data collected during stationary periods (defined as movement speed <1 cm/s) to assess their performance during these stationary periods.

Random-select cells: to ensure fairness of comparison, we performed the same trajectory decoding process using randomly selected cells after removing all hand position-tuned cells from the population of all cells (random-select cells1) or stable cells (random-select cells2). For example, suppose we have identified $n$ primary hand position-tuned cells within a specific session. Subsequently, we randomly selected $n$ cells from the remaining cells from the same session and evaluated their decoding accuracy. This process was repeated 100 times for each session and the average accuracy was calculated as the random-select result.

### Reporting summary

Further information on research design is available in the Nature Portfolio Reporting Summary linked to this article.

## Data availability

The processed data used to generate results in this study are provided in the Source Data file. The raw electrophysiological data used for this study are available from the corresponding author upon reasonable request. Source data are provided with this paper.

## Code availability

Custom code associated with this study is available at https://github. com/caoshenghao/pmd_position_cell.

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

## Acknowledgements

We thank Dong-Lin Gu, Peng-Cheng Pan, Kai-Xi Tian, Cheng-Teng Jiang, and Jing-Wen Guo for technical assistance during the animal experiments. We thank Wen Ren for helping with setting up the camera systems. We appreciate the professional care provided to the animals by Yan-Yan Liu and Bao-Jiang Niu. We thank Rui Chen for helping draw the diagram of the behavioral task. We thank Hong-Dian Yang, Cheng-Lin Miao, Da-Jun Xing, and He Cui for their comments on the manuscript. This work was supported by the STI 2030—Major Project 2021ZD0200402 to S.Y., and 2021ZD0200200 to T.Z.J.

## Author contributions

S.C. and S.Y. conceived and designed the study. S.C., X.H., and Z.Z. performed the implantation surgeries. S.C. and X.H. performed the behavioral experiments and data recording. S.C. and Z.Z. preprocessed the neural recordings. S.C. wrote the analysis code, analyzed data, and visualized results. All authors interpreted and discussed results. S.C. and S.Y. wrote the initial manuscript. S.C., S.Y., J.W.G., and T.Z.J. revised the manuscript.

## Competing interests

The authors declare no competing interests.
