## [Peer Review file · Nature Communications]

Hand Position Fields of Neurons in the Premotor Cortex of Macaques during Natural Reaching

Corresponding Author: Professor Shan Yu

Version 0:

Reviewer comments:

Reviewer #1

(Remarks to the Author)

In this manuscript the authors explored the presence of neurons with place-field-like tuning for hand position during natural reach-and-grasp movements in the dorsal premotor cortex of monkeys. Their main conclusion, as stated in abstract, is that “cells in the dorsal premotor cortex (PMd) of macaques exhibit field tuning of hand position during a natural reach-and-grasp task, similar to hippocampal place cells”. The manuscript is well written, and the results are presented clearly, but the study do not really provide a strong advance on what we already know about the properties of PMd neurons.

The strength of the study would have been a description of PMd activity during reaching movements in a 3D space. The lack of the Z axis in most of the analysis, however, did not allow for a 3D spatial representation. The absence of the Z axis is a major limitation.

It might be interesting for the authors to investigate the preferred space of neurons in terms of a tuning curve, similarly to the preferred direction during center-out reaching tasks. Then, relating the spatial preference of neurons to their locations in the array would allow to test presence of a spatial map within the PMd. But, again, without the Z axis it would be only a 2D map.

Fig 2b should be presented for all monkeys.

The discussion in its current form looks like a short abstract/summary, not a real discussion, and it needs to be expanded.

Reviewer #2

(Remarks to the Author)

Spatial map in Premotor Cortex

In this interesting study, the authors argue that besides tuning for various variables such as speed and hand movement direction, the dorsal premotor cortex also codes the location of the hand and that it is similar to hippocampus in its representation. The authors argue that this is an analogue of “place cells” and also suggest the presence of grid and border cells. The manuscript is written well and compact and the analyses clearly performed.

I think the premise is interesting but I am a little unsure about whether this is just yet another example of how a high-level premotor area that is involved in the planning and control of movement will have many variables that the neurons will respond to. I think principally the question is one of effect size and how strong the effect is which I explain below. When reading the paper I was worried that the neuron responses were due to eye position tuning and also the proprioceptive/EMG in that location since we know that PMd and M1 have signals associated with EMG as well. So I would need to see a clearer demonstration that it is not other confounding factors that lead to the effects that they see!

I sympathize greatly with the authors but said differently this is the age-old problem of kinetics vs. kinematics and what the neurons in premotor and motor cortices are “tuned” to. Ultimately, I expect the FR of a PMd neuron is a nonlinear combination of the hand speed, reward location, hand direction, eye position, EMG etc and the authors need to provide a relative effect size for each of these variables. Then hand position might be one of the many parameters that influence the

results and not since these variables are correlated with one another, there is a chicken and egg problem here.

Major Comments:

1. I think the difficulty I have with the claim that the PMd has a spatial map of hand position is that it is but one of the many variables in this brain area. As the authors themselves acknowledge there are certainly other factors that could matter: hand speed, location of reward, direction of hand. I think the authors have done ok in assessing some of the covariates but a better assessment of how reward location affects the responses, and two other covariates need to be addressed to be fully convincing. In slightly more details:

a) What are the eye movements of the monkey during this task? Even if the authors don't have the eye movements, even an assessment of the face or some coarse measure of gaze direction would be important to rule out as a confound. The reason I am saying this is that in monkey X and also somewhat in other monkeys they are in the rostral PMd and this is known to be an eye movement related region that is close to the FEF. I think if the authors showed the hand position tuning more caudally then that would be less of an issue but PMdr is definitely a hotspot for eye movements.

b) How do the authors rule out that this is not a proprioceptive signal or a causal signal to hold the muscle in a particular position.

c) I notice even in the first and second examples, there is a hotspot of firing near the mouth of the animal. I was wondering whether some of the responses are due to peripersonal space/arm reaching space. There is also this notion of the arm reaching space as well in these brain areas. This leads to additional consequences for interpreting the results that the authors propose that this is a spatial encoding of hand position.

d) The assessment of the effect of reward needs to be quantified especially given that the supplementary figure currently just shows a few reward locations ("many did not correspond to the location where the reward was frequently given (Supplementary Fig. 8)"). Some statistical assessment of first regressing out reward location and then performing the place field analysis is the right way to do it. In fact the general approach I would take is to first regress the firing rate of the neuron as

$FR = f(\text{reward location, eye position, hand speed, hand direction})$ and then use the residuals and examine if they have place field encoding. Such an analysis is necessary in my opinion to claim the existence of a place field like tuning in the premotor cortex for hand position. I think such an analysis would assuage all the issues in major comment 1

Minor Comments:

1. I noticed some of the arrays are more in PMd and others are more in PMdr. Are there differences between the amount of hand position tuning in PMdr compared to PMd

2. Figure 1d is very different from figure 1c. I would like to see a heatmap of Figure 1c plotted in Figure 1d with minimal post processing. Why is the hotspot near the mouth for instance not visible in Fig. 1d

Reviewer #3

(Remarks to the Author)

The authors have presented a novel and intriguing finding of the activity and representations that can be found within the dorsal premotor cortex (DPC). In particular, using a simplified reaching task, they found that certain cells in the DPC encode the physical position of the hand, described as "position cells". Further, the position cells that were identified were, in some cases, completely separate from representations of velocity and direction that have been reported previously. The size, location, and shape of the position fields associated with each position cell were characterized, and the presence of other spatial encoding cell types was tested. Summarily, while some other types of spatial encoding cells were found in minor quantities, the spatial map encoded in the activity of the DPC is entirely attributed to the position cells.

While these findings are very promising for advancing our understanding of the role of this premotor cortex, there are a variety of technical issues that must be resolved before considering this manuscript for acceptance.

Major Changes:

1) In the current version of the manuscript, there appears to be a notable lack of technical control over the parameters of the task, as well as a lack of a reasonable and relevant control task. Please address the following questions in order to clarify

the framework of the study.

- a) The manuscript states that 100 trials are used per recording session. How many unique positions occur within the 100 trials?
 - b) What locations were maintained constant between different monkeys, or between different sessions for the same monkey?
 - c) How many trials are presented for each unique location?
 - d) Are all the positions maintained at exactly the same height (Z-dimension)?
 - e) If the locations are largely non-overlapping, how could unified conclusions be drawn from mismatched sets of locations?
 - f) Supplementary figure 8 and its caption suggests that there are few (2 or 3) positions for grasping. Such a characteristic does not agree with the mapping in the movement.
 - i) Please be more detailed and add a supplementary figure with the spatial distribution of targets in the XY, XZ, and YZ planes to better elucidate the framework of the reaching task.
 - ii) What purpose does isolating these few positions serve in developing the manuscript's narrative?
- 2) Beyond the parameters of the task itself, the authors have failed to specify how the video recordings were coordinated with activity patterns from the recorded neurons. How was video synchronized to neural activity? This detail is critical due to the fact that most programs need a common event recorded in both signals to use as a reference.
- a) Additionally, a time jitter will occur with such a great mismatch between sampling frequencies, which the authors should have accounted for, although this procedure has also been omitted from the manuscript.
- 3) Related with the previous point. In Figure 1, the reach trajectory graphs (Fig. 1C) seem to lack some corresponding representation when compared against the spatial density plots (Fig. 1D). I have included 3 concrete examples that should be individually addressed:
- a) Cell 43 (Fig. 1C, left) has the greatest density of spikes very close to the "nose" location of the west section. Further, there is clearly observable activity in the northeast section. Why are these representations missing from the corresponding spatial plane that was generated (Fig. 1D, left)?
 - b) Cell 21 responds strongly near the nose, both below and above it, with a density of spikes that appears to be the most prominent across all the reach trajectories. The west and northwest sections of Fig. 1D (middle-left) are lacking any visible indication of these dense areas.
 - c) Cell 3 responds with the greatest density in the south section of the plane (Fig. 1C, right), although this is missing from the spatial map (Fig. 1D, right).
 - d) Please clarify why these results are consistent between different cells when the visualized trajectories do not coincide between the 4 visualized cells?
- 4) While the authors have made an effort to characterize the heterogeneity in the recorded DPC population, their process for sub-selecting groups with relevant and similar dynamics could be biasing the observed results.
- a) It is important to apply a population dynamics analysis to the general, trial-averaged activity of all the neurons, not just the "relevant" ones
 - b) Further, the idea of a spatial map should be a property that is intrinsically associated to the individual neurons, as the authors have claimed in the present work. Based on this conclusion, it is necessary to apply an analysis of population dynamics based on the individual trials, which could be feasibly computed using the Latent Factor Analysis via Dynamical Systems (LFADS) metric. For examples of implementation of this metric, see Pandarinath et. al, 2018 (Nature Methods).
 - c) How do the dynamics of individual trials relate to the general population dynamics from trial-averaged activity?
 - d) The population dynamics should be useful in characterizing the spatial map in a more compact format, which would aid the interpretability of Fig. 1, 2, 3, although especially so for Fig. 1C
- 5) The decoding of varying physical variables related to the reaching task in Fig. 4 is not sufficiently justified with references, and it is difficult to follow the analysis as it is in the current state. Maintaining a focus on the individual units is a glaring oversight, since the authors have clearly identified both separate, and common, subpopulations that can be studied as independent entities. Given that the populations representing direction and velocity are separated from the population representing the position, how can the authors determine that the position-tuned cells are not integrating velocity and direction signals, since these would suffice to pinpoint the physical location?
- a) Please apply a separate analysis of population dynamics to this particular section of the study, since this could provide far stronger evidence for the isolated functionality of the purportedly separate populations.
 - b) It is also essential for the authors to justify the metric used for the reconstructions, preferably using previous examples from recent work.
- 6) Recent work in the DPC has shown that this area integrates both somatosensory information, as well as information from the visual sensory modality. How can the authors know that the cellular activity is related to the physical position of the hand, and not a trace related to integration of visual information?
- a) Have the authors recorded data in any primary visual areas or somatosensory areas?
 - b) Does some control exist that can address this question? If so, this control needs to be highlighted, and if not, the authors should describe what kind of control scheme could allow for these doubts to be addressed.
- 7) The decoder that is used in Figure 2 needs to be explained in greater detail. The populations of position cells and all cells need to be stated explicitly, including mentioning how the decoder's performance may change based on the differences between the response properties of the two separate populations. How are these two populations related to the populations indicated in Fig. 4?
- a) If all the cells can be used to decode the position, then why are the authors focused only on the position cells?

b) This decoder should be applied to the various subpopulations that have been described (839 putative single-units, 609 single-units with stable XY-spatial firing rate maps, 132 cells meeting the desired criteria)

8) In Fig. 2B, the X-centered trajectories and the Y-centered trajectories have vastly different scales (X: -20cm - 20cm; Y: -100cm - 100cm). Based on this, how can the X-graph (Fig. 2B, Top) be compared to the Y-graph (Fig. 2B, Bottom). Furthermore, the observed Y-centered trajectory diverges in many instances from the decoded trajectory from both populations of cells, which could be indicating that the information related to the Y-centered position is being contaminated by slight variations in the Z-dimension.

a) I would like to ask the authors to apply the same decoder to static periods of no movement, as a control to demonstrate the decoder's efficacy

b) I would also like to recommend calculating the same graph for the Z-centered position

c) Can the authors identify some reason in particular that could explain the discrepancy between decoding X- vs. Y-centered positions?

9) It is unclear how properties that are included about the position fields of the cells (Fig. 3) are related to decoding the position.

a) What significance does an ovalar position field hold to the overall conclusion of the article?

b) The graphs for spatial sparsity and spatial coherence are so similar, so why include both? What benefit comes from having both?

c) What is the relationship between position field size/shape and sparsity/coherence?

d) Which are the optimal properties to decode the position fields?

10) The current manuscript has focused on analyzing the recorded populations of neurons in the DPC, although their treatment on the subject is severely lacking with respect to population dynamics and heterogeneity observed across the population.

a) Reflect on Kaufman et. al, 2014 (Nature Neuroscience); this article frames the activity in DPC based on a limited gating of preparatory activity that permits isolation of execution dynamics. It is important to discuss how the spatial maps could interact with preparatory activity.

b) Recent work has identified mixed selectivity and associative activity patterns in the DPC. Please comment on the relationship between the spatial map and the decision-making frameworks that have been presented (Elsayed et. al, 2016, Nature Communications; Vyas et. al, 2020, Neuron)

c) Other works have shown that the DPC (PMd) plays a significant role in other cognitive processes related to abstraction (Diaz-deLeon et. al, 2022, PNAS), execution of movement (Churchland et. al, 2012, Nature; Wang et. al, 2019, Nature Communications; Glaser et. al, 2018, Nature Communications), and maintenance of decision signals (Rossi-Pool et. al, 2017, Neuron; Ohbayashi et. al, 2003, Science)

d) How does a spatial map and representation interact with the different signals and heterogeneity that has been described in recent work?

11) The activity observed in DPC can be related to the activity that has been previously described in the hippocampus. The authors mention this point briefly, although it should be discussed with more care and depth.

a) What traits of the activity in DPC are directly relatable to the traits of the activity in the hippocampus?

b) Are there properties of the activity observed in DPC that are completely distinct from those observed in the hippocampus? If so, what are they, and what does their presence indicate about the conclusions that the authors have reached?

c) Has the hippocampus been studied in this task? Has it been studied in any similar tasks? If not, how can the authors claim any association between the activity in DPC and hippocampus?

12) The discussion section of the current manuscript is lacking in its consideration of the novelties presented recently in the field, and it does not place the presented results in an adequate context for understanding how the observed results are related to the general body of work done in DPC, as well as the body of work dedicated to reaching tasks.

a) Reflect on the context of DPC and how the results can be considered part of, or contradicting, the previously considered models

b) It is essential to return to the consideration of place and grid cells, and how they relate to the putative "hand-place" cells that were observed in DPC

c) Reflect on the relationship between population dynamics and the spatial map. Is the spatial map solely represented in "position" cells? Can spatial maps be decoded from other, place-ignorant cell subpopulations?

d) Consider mentioning Graziano, Taylor & Moore (2002, Neuron) (this point is not important at all)

13) The Pearson correlation coefficients that are employed for calculating the speed score, the grid score, and the recording stability have no mention of the significance criterion used. This oversight is compounded by the lack of false positive validation. In the case of the recording stability the halved pieces of the dataset should be correlated with the full dataset, in order to have a reference frame for the correlation coefficient that is obtained.

Minor Changes:

1) The introduction section is limited in length and scope, which greatly decreases the interpretability of the manuscript as a whole. Along with the corrections previously mentioned, please summarize the general results of the manuscript as the final paragraph of the introduction.

2) The legend for Fig. S2 is entirely lacking, with no description of what is contained within the figure. Please focus on describing the differences between each of the panels, and the significance of including them for the resulting conclusions.

3) Mutual information estimation usually needs a correction due to the finite sampling method, especially when considering the fact that the underlying distributions need to be well estimated. A variety of shuffling techniques have been proposed to address this particular problem, although the authors make no mention of such a procedure.

a) How many trials were used to calculate the spatial information related to each individual location?

b) Please describe how distributions were estimated, and if/how these distributions varied by neuron, monkey, or session.

4) The cells that are used as examples in Fig. 1C & D are abandoned for a 5th, unrelated cell in Fig. 1E & F. Further, this 5th cell is not from any of the recording sessions that were included previously. This discrepancy may be understood as suspect, and it could even give the impression that only the X and Y dimensions were measured for the first 4 neurons.

5) The authors have purportedly shown that cells in the DPC maintain information about the location ("place") of the hand in 3-dimensional space. They proceed to analyze whether these same cells have qualities related to grid or border cells, but the authors fail to mention what the relevance of identifying grid cells or border cells is to the primary finding of the manuscript. Without this additional information, the last section of results holds almost no bearing on the general conclusions that are reached.

6) The authors have focused their analyses on analyzing the firing rate from the relevant group of neurons, although they have omitted the parameter details used to calculate the firing rate as a function of time.

a) These details are essential to understanding the study as a whole, so they must be specified.

b) The activity rates of different neurons vary greatly. How can these different rates of activity be used to compare single-units? One common solution is to normalize the rates of activity using a transform (e.g. z-score transform), although this procedure is not mentioned anywhere in the manuscript.

7) In lines 136-137, the value is denoted as larger than the expected value by chance, so please include what the chance value should be. The same comment holds true for lines 139-142.

8) The word "grid" is misspelled on line 285

Reviewer #4

(Remarks to the Author)

In this manuscript, the authors examined whether cells in the dorsal premotor cortex (PMd) of macaque monkeys exhibit field tuning of hand position during a natural reach and-grasp task. To test the hypothesis, the monkeys were trained to reach and grasp fruit pieces from a wand held by the experimenter. While the monkey was performing the natural reach-and-grasp task, the authors recorded activity of PMd neurons and the 3D position of the right hand using 4 video cameras. Their analysis of activity during movements showed that a sizable proportion of PMd neurons exhibit significant spatial specificity of firing. A proportion of PMd neurons exhibited selectivity for hand moving direction (51%) and speed (24%). The authors concluded that their findings suggest that the PMd may utilize a similar encoding framework as the hippocampus to guide goal-directed limb movements.

Testing to see if the similar computational mechanisms as in the hippocampus exists in PMd is an unique attempt. The study analyzed data during the movements. In addition, the target fruit location was changed to a new position while the monkeys were reaching in some trials. These left whether the sustained activity before the movements also has similar spatial specificity as an unanswered question.

In the abstract and introduction, the authors stated about the hippocampus (line 6, 12, 16, lines 19-31) more than the PMd (line 11, 15, lines 32-40) even though they did not do experiments in the hippocampus. Some readers may expect to see the comparison of data between the PMd and hippocampus. Please balance the proportion to reflect the actual experiments and results.

Locations of implanted array electrodes in Figure 1a makes an impression that some data may be recorded in M1 or PMdr. For example, in Monkey X, a part of the implanted array looks like in PMdr (rostral PMd). In monkey B, a part of the array looks like in M1. The results must be different between the PMdc, PMdr and M1. Would you add central sulcus to each map to clear this point?

In Figure 1c, what each red dot represents is not clearly written. Please write it in the method section.

Version 1:

Reviewer comments:

Reviewer #1

(Remarks to the Author)

Overall, I think the authors did a nice job in re-writing the whole discussion and in addressing some of my points. As I stated

for the previous version, I think the manuscript is well-written, but honestly, I still do not see how this study would advance what we know about the premotor/motor spatial encoding of movements.

For example, among important and relevant works, early studies showed that M1 neurons encode spatial gradients when holding the hand at specific orientations in a 2-D space (Georgopoulos et al., 1984, *Experimental Brain Research* (54)446–454) as well as in a 3D-space (Kettner et al., 1988, *J Neurosci* (8)2938-47). These studies were later followed by the two Caminiti's papers (Caminiti et al., 1990, *J Neurosci*, (10)2039-2058; Caminiti et al., 1991, *J Neurosci*, (5)1182-1197) in which the authors clearly demonstrated that both premotor and motor cortices, beyond their coding of movement direction, have an internal representation of space for the arm movement (it is interesting to note that they did not use the typical center-out paradigm).

I feel the present study is a more "naturalistic version" (which is of course welcome and needed) of old studies, but it does not advance our knowledge of premotor/motor areas spatial encoding of arm movements.

Reviewer #2

(Remarks to the Author)

I apologize for the delay in returning the manuscript. The start of fall with attendant challenges made me delay my review. The authors have tried to address the comments of the reviewers with many different analyses. So that is a positive.

However, I unfortunately must say that I still remain unconvinced by some of the key analyses. To claim that there is a unique hand position tuned population of cells in PMd like the hippocampus needs a level of evidence that unfortunately is not present in the manuscript at this point. Based on the data provided, I cannot in good conscience recommend this paper for publication. I am not sure the 50 they find are uniquely hand position tuned because of the following problems. Note, this is one thing I dug into because I feel this was the one case where they performed analyses to address my comments and it was unsatisfactory.

The point is ideally illustrated by the regression analysis. I might be taking an extreme position, but I fear "tuning" in the premotor cortex might be dead on conception. Basically, their analyses themselves reveal that because of the correlation between hand-speed, hand-direction, reward location, and eye position, they all are explanatory of the data. So when you control for one variable, the others explain very little variance suggesting that the representation perhaps is not as simple as one thinks. I don't even fully believe that there are unique hand direction cells in older experiments, they are just one of the many variables that experimenters tested. So if you really want to make these claims, you need to experimentally decouple some of them. There are researchers who have done brutally hard experiments, where they make monkeys fixate in one location while reaching to the other to decouple eye centered from hand centered reference frames, or reach in 3d using Wheatstone bridges and 100s of trials and even in those cases, the results are super complicated. The experiments in the paper lack such controls and thus the results are muddy.

The authors perhaps might argue that I am being too conservative in my thinking and should go purely by the reconstruction analysis proposed. Even if one sets aside the regression analysis there are huge problems with the claims. Unlike the bat data, which is highly skewed outside the unity line, in the PMd data, the distribution looks symmetric around the unity line (compare Fig. 3c, d and the figure in the responses to the reviewers Fig. 10a). So the authors basically take stuff outside the unity line from Fig. 3d in the position vs. hand-speed variable based reconstruction analysis. But when they look at the data using reward location basically everything falls on the unity line with some symmetric spread around it. In this case, would you not need to correct for multiple comparisons here and also assign a p-value for each cell based on a shuffle test? Doing so might reduce the number of "place cells" in the population even more. So overall even this analysis is unconvincing. And given that there are no eye movement controls, the 50 cells become even more suspect in my opinion.

In addition, given that these are fixed Utah arrays pooled across days, the number of units is also not unique. Maybe the same units are multiple counted on different days, which would reduce from 50 to even less. So after all these corrections, if we end up with say 3% of hand position tuned cells, is this biologically meaningful? In the bat data: All 58 of their cells were goal tuned and NOT place tuned.

Perhaps, the authors want to say, see how good our decoding analyses are from these 50 cells. However, this has problems as well. Did the authors compare the decoding correlation for a random 50 cells from their population and show that the resulting correlations are different? A shuffle control is meaningless. Take a random 50 cells from the stable cells and repeat and show that the distributions are fully different. Such an analysis is necessary for the claims in the manuscript. They might have done this somewhere and I missed it. But the way the authors have written the paper with so many supplements and just 3 main figures that are sparsely described in figure legends makes it all impossible to follow.

Overall, I am unsure this manuscript clears the bar for the claims in the manuscript. Let aside questions of novelty, impact etc. I just cannot be sure the experimental control and statistics support the conclusions in the manuscript. There may indeed be hand position tuning in PMd, but this dataset and analyses do not meet that bar.

I have attached a PDF with relevant figures from the manuscript and response highlighting my confusion and difficulty with the claims?

P.S. Both the manuscript and responses to reviewers are riddled with typos, that make following it hard.

P.P.S I do apologize if the tone in my review feels strident. But I wanted to voice my worries about the data in as clear a manner for the authors as possible and note everything cannot be a discussion point that will be addressed in future work!

Reviewer #3

(Remarks to the Author)

The authors have presented a strong narrative for the hand-position coding that can be observed in the dorsal premotor cortex (DPC). Furthermore, this manuscript contributes to a newly emerging body of knowledge that revolves around studying the kinds of signals that occur in behaviors that are more suited to the monkeys' natural environment. While this regime of studies is different from the classical task-based framework, the value it holds in addressing unanswered questions is apparent from the current work. Although I initially believed that the manuscript required a considerable amount of revision in order to be ready for publication, the authors have gone above and beyond in their responses and consideration of my commentary. As such, I would like to wholeheartedly support the publication of this submission in its current form.

Reviewer #4

(Remarks to the Author)

The authors have adequately addressed my concerns from the initial review.

Version 2:

Reviewer comments:

Reviewer #1

(Remarks to the Author)

This manuscript has undergone significant changes since the first submission. I think the authors have done a good job reframing the manuscript to be more suitable and coherent, and they have addressed my previous concerns. Upon re-reading the latest version, I have only two points that I think should be revised or clarified:

1) Perhaps I missed this in the previous versions, but the authors state, "we collected 839 (Monkey A, 51; Monkey B, 101; Monkey X, 390; and Monkey Z, 297) putative single units (SUs)." However, the analysis was conducted on 601 neurons. The selection process leading to this reduction is unclear.

2) In the discussion, it is not clear how the authors conclude, "we argue that the field-like hand position coding in the PMd is egocentric and would remain constant after body relocation, which is distinct from the allocentric space representation for body navigation."

(Remarks on code availability)

Reviewer #2

(Remarks to the Author)

I like the revisions and the reframing of the paper and it feels less dramatic as a claim. I think given the 4 monkeys worth of data and a major softening of the claims of the paper make it more suitable for publication now. I also like the natural behavior.

However a couple of other recommendations just to tighten the results and this could be reviewed by the editor and included in supp. information.

1) The authors should perform a neuron dropping analysis to take all the hand position tuned neurons and remove them from the decoding analysis in Fig. 3H for the all cells and stable cell population. If the decoding accuracy drops sharply that supports their claims even more. I mean even a random set is quite good, the performance is reasonable. To strengthen their claim, show the random reconstruction in a separate panel in Figure 3G perhaps with the position cells to bolster this claim.

2) The authors should discuss the effects of eye movements especially given that some of the arrays are close to the arcuate sulcus.

3) Are these hand position tuned neurons spatially organized on their arrays?

4) All in all this is interesting because it uses natural hand tasks to discover position tuning. However, this tuning might be an artifact of the need for a dynamical system to have representations. This might be worth discussing as well as an alternative to their Tolman Eichenbaum machine. As the authors themselves show there ARE a lot of variables being represented. Work from Churchland et al. 2010, and Churchland et al. 2012, Lara et al., Jazayeri lab papers etc, Sussillo et al. 2015, Boucher et al. 2023 etc might be relevant for this contrast.

(Remarks on code availability)

REVIEWER COMMENTS AND REPLY

Reviewer #1	page 1 - 5
Reviewer #2	page 6 – 16
Reviewer #3	page 17 – 41
Reviewer #4	page 42 – 43

Note: We have adjusted many contents of figures according to the reviewers' comments. We annotated the new figure numbers (***bold, italics***) corresponding to the original figure numbers in the reviewer's comments, and we only refer to the new figure numbers in our Reply.

Reviewer #1

In this manuscript the authors explored the presence of neurons with place-field-like tuning for hand position during natural reach-and-grasp movements in the dorsal premotor cortex of monkeys. Their main conclusion, as stated in abstract, is that “cells in the dorsal premotor cortex (PMd) of macaques exhibit field tuning of hand position during a natural reach-and-grasp task, similar to hippocampal place cells”.

Comment 1: The manuscript is well written, and the results are presented clearly, but the study do not really provide a strong advance on what we already know about the properties of PMd neurons.

Reply 1:

We appreciate the reviewer's comments. The main finding, as the reviewer correctly pointed out, is the field tuning of hand position in PMd during a natural reach-and-grasp task, which demonstrates previously unknown characteristics of PMd neuron in representing spatial information.

Previous studies based on e.g., the center-out paradigm, did not succeeded in delineating the selectivity for movement direction and target position. Other studies reported that the hand position relative to e.g. reach targets can influence PMd neurons' responses[1], [2], but they did not reveal the nature of such position tuning. Intriguingly, despite its close association with movement direction and speed, our findings underscored the surprising necessity for a direct representation of hand position, even within such simple, spontaneous reaching tasks. Thus, the discovery of the field tuning of hand position in the present study shed important new light on how the spatial information is encoded in the PMd. We have incorporated additional content to provide a more detailed discussion of these aspects. (**main text: line 155-160, 198-211**)

Comment 2: The strength of the study would have been a description of PMd activity during reaching movements in a 3D space. The lack of the Z axis in most of the

analysis, however, did not allow for a 3D spatial representation. The absence of the Z axis is a major limitation.

Reply 2:

We agree that it would be important extension of the current results to the 3D space. To provide information regarding the tuning in 3D, we have provided the 2D results in all three planes, i.e., the XY, YZ and XZ plane, which exhibit highly similar field tuning properties (**Supplementary Fig. 11**). We also directly analyzed neuron's response properties when 3D hand positions are taken into account. Overall, the firing properties are similar to the field tuning that we characterized in 2D. Some examples of 3D tuning are shown in **Fig. 1e** and **Supplementary Fig. 4**. Further, we pooled response profile of all neurons recorded in individual sessions together and found that their field-like tuning cover significant proportion of the 3D space actually explored by the animals (**Reply Fig. 1**). These results provide evidence that our main findings regarding the field tuning of hand position in the PMd may well extended to 3D condition.

However, as we explained in the text, to fully examine the 3D field tuning requires much larger sampling space and therefore, significantly longer recording per session. We are currently designing new experiment to address this issue properly. In the revision, we have acknowledged the limitation of lacking comprehensive characteristics of field tuning in 3D, and discussed future work needs to address it (**main text: line 170-174**).

Reply Fig. 1. Distributions of position fields in 3D space. We visualize position

fields from putative position cells as convex hulls. Colors distinguish fields from different cells. The gray meshes indicate the envelope of areas the hand passed through during the session. Four exemplar sessions (one for each monkey) are shown. We note that in order to calculate the 3D position field based on limited data, we need to loose the stringent criteria used in the 2D condition, e.g., change the definition of the field border to be $>$ than 30% of the maximal firing rate, instead of 50% and cancel the requirement of minimal volume size or number of visited trajectories.

Comment 3: It might be interesting for the authors to investigate the preferred space of neurons in terms of a tuning curve, similarly to the preferred direction during center-out reaching tasks. Then, relating the spatial preference of neurons to their locations in the array would allow to test presence of a spatial map within the PMd. But, again, without the Z axis it would be only a 2D map.

Reply 3:

We thank the suggestion of investigating the preferred space of neurons in terms of a tuning curve. In center-out reaching tasks, the spatial degree of freedom (DOF) is one--the direction of movement and the tuning curve is a function that describes the relationship between the firing rate of a neuron and the direction of movement. In our task, there are two spatial DOFs (or three if you consider 3D space). Our analyses were based on the spatial firing rate maps which is actually a 2D version of spatial "tuning curve". The preferred space of neurons is indeed what we called "firing field" in the current study.

As the reviewer suggested, to examine if there is a spatial map, i.e., a continuous field position distribution, within the PMd would be an interesting question. Grazinon et al has reported that microsimulation of PMd neurons elicit arm movements to specific spatial location, but the final location did not appear to be mapped across the cortical surface[3]. In our data, to address this issue, we choose two monkeys' data that we can determine the location of individual electrodes on the cortical surface (monkey B and Z), and analyzed the relationship between the spatial preference of neurons and their locations in the electrode array: we plotted the distance between centroids of the firing field as the spatial preference versus the distance between the positions of the electrode from which the neurons were recorded (see **Reply Fig. 2**). We found no significant correlation.

In addition, also for monkeys B and Z, we placed the spatial preferences of neurons in the electrode array (see **Reply Fig. 3**) to see if the neurons with similar spatial preference are clustered in the array. Due to the limited cortical area covered by the electrode array and the limited number of neurons with spatial preferences recorded, we found that the current results are inconclusive.

In summary, our non-positive results may reflect the lack of spatial map, as suggested by previous microsimulation study. But they may also be due to other factors such as insufficient data. We plan to address this question in future studies in which we can collect more comprehensive datasets. At this point, we have added a

discussion of this issue to the paper (main text: line 233-234).

We modified the title to 'A Map of Hand Position in the Premotor Cortex of Macaques' to avoid any misunderstanding that our study comprehensively covered the arrangement of spatial preferences in the PMd cortex.

Reply Fig. 2. The relationship between field distances and neuron distances. Blue dots indicate neuron pairs. The red line indicates the result of the linear fit. Neurons with only one field were included in the analysis.

Reply Fig. 3. Spatial preferences of the horizontal (XY) plane within the electrode arrays for two monkeys. Colors indicate the field positions referred to the monkey's nose ($x=0$, $y=0$). Red and green indicate left and right respectively. Higher brightness and lower saturation represent closer positions to the monkey; lower brightness and higher saturation represent further positions.

Comment 4: Fig 2b (**Fig. 3g**) should be presented for all monkeys.

Reply 4:

Thanks for the suggestion, we have added the results of the other monkeys in

Supplementary Fig. 10, and we have also added the trajectories of Z axis (calculated from the XZ plane) in both **Fig. 3g and Supplementary Fig. 10**.

Comment 5: The discussion in its current form looks like a short abstract/summary, not a real discussion, and it needs to be expanded.

Reply 5:

Agreed! We have re-written the whole **Discussion** section to cover related issues including the following key points: 1) how we control other possible interference from muscle signals, peripersonal space representation, and eye movements; 2) the relation between position tuning and population dynamics; 3) the potential origins of position signals; 4) the region-related variabilities of the hand position tuning; 5) the relation between position tuning and other functions of the PMd and etc.

(main text: line 155-274)

Reviewer #2

Spatial map in Premotor Cortex

In this interesting study, the authors argue that besides tuning for various variables such as speed and hand movement direction, the dorsal premotor cortex also codes the location of the hand and that it is similar to hippocampus in its representation. The authors argue that this is an analogue of “place cells” and also suggest the presence of grid and border cells. The manuscript is written well and compact and the analyses clearly performed.

I think the premise is interesting but I am a little unsure about whether this is just yet another example of how a high-level premotor area that is involved in the planning and control of movement will have many variables that the neurons will respond to. I think principally the question is one of effect size and how strong the effect is which I explain below. When reading the paper I was worried that the neuron responses were due to eye position tuning and also the proprioceptive/EMG in that location since we know that PMd and M1 have signals associated with EMG as well. So I would need to see a clearer demonstration that it is not other confounding factors that lead to the effects that they see!

I sympathize greatly with the authors but said differently this is the age-old problem of kinetics vs. kinematics and what the neurons in premotor and motor cortices are “tuned” to. Ultimately, I expect the FR of a PMd neuron is a nonlinear combination of the hand speed, reward location, hand direction, eye position, EMG etc and the authors need to provide a relative effect size for each of these variables. Then hand position might be one of the many parameters that influence the results and not since these variables are correlated with one another, there is a chicken and egg problem here.

Reply:

We appreciate the overall positive evaluation and helpful comments regarding the relation between the tuning for hand position and other tuning properties in the PMd. We have implemented new analyses to better address this issue. Please see below for details.

Major Comments:

Comment 1: I think the difficulty I have with the claim that the PMd has a spatial map of hand position is that it is but one of the many variables in this brain area. As the authors themselves acknowledge there are certainly other factors that could matter: hand speed, location of reward, direction of hand. I think the authors have done ok in assessing some of the covariates but a better assessment of how reward location affects the responses, and two other covariates need to be addressed to be fully

convincing. In slightly more details:

Comment 1 a): What are the eye movements of the monkey during this task? Even if the authors don't have the eye movements, even an assessment of the face or some coarse measure of gaze direction would be important to rule out as a confound. The reason I am saying this is that in monkey X and also somewhat in other monkeys they are in the rostral PMd and this is known to be an eye movement related region that is close to the FEF. I think if the authors showed the hand position tuning more caudally then that would be less of an issue but PMdr is definitely a hotspot for eye movements.

Reply 1 a):

We agree that eye movement is an important interference factor to be consider. Unfortunately, in this study, we did not record eye movement data during the experiments.

According to the suggestion, we first reviewed the recorded videos in which the eyes, the hands, and the rewards could be seen. The usual pattern we observed was that when the food appeared in sight, monkeys immediately looked at the food, not their hands. When there was no food in sight, the monkey's gaze direction was more or less random, and its hand was often put somewhere on the primate chair. We attached some video clips to illustrate such pattern (**Reply Video 1**). The lack of correlation between gaze direction and hand position suggests that the tuning of the latter we observed is less likely be affected by eye movements.

Second, as mentioned by this Reviewer and also Reviewer #4, our implantation sites in PMd varied among four monkeys. The fine brain regions covered by the electrode arrays might contain most of PMd/PMdc and a small part of PMdr, PMv, and even M1. Our monkey-by-monkey analysis in **Supplementary Fig. 12** shows that this hand position tuning is common and existed outside PMdr. So, although we cannot fully exclude the possibility that eye movements may affected some of the neuron activity recorded in monkey X, it is unlikely that such inference will strongly affect our overall results (**main text: line 221-230, 234-240**).

In the discussion, we have added a part to delineating possible interfering factors including the eye movement (**main text: line 234-240**). We have also designed further experiments to include eye movement recordings during the reach-and grasp task, which will be able to address this issue in a more quantitatively way. Thanks again for this helpful suggestion.

Comment 1 b): How do the authors rule out that this is not a proprioceptive signal or a causal signal to hold the muscle in a particular position.

Reply 1 b):

Thanks for the question. We think that the issue related to proprioceptive signals is more a question of the mechanism underlying the field tuning that needs to be addressed in the future, while the interference related to muscle control signals can be largely excluded based on our data and a precious study.

First, we note that the field tuning has to rely on the information regarding the coordinated position of hands in space, which could be inferred from upstream visual or proprioceptive signals[4]. The ability to reach and grasp when we close the eyes suggest that the proprioceptive signals may indeed play an important role here. However, it is related to the mechanism that gives rise to the field tuning we observed, rather than a confounding factor that casts doubt on the validity of the tuning itself. In the future, we plan to carry out experiments for example in totally dark environment to better delineate the influence of visual and proprioceptive signals in forming the position fields in the PMd. **(main text: line 212-220)**

Regarding the possibility that a causal signal to hold the muscle in a particular position may contribute to the field tuning, we note that for this to happen, the moving direction and speed when the arm passing the particular position needs to be the same or at least very similar, otherwise the signal to hold the muscles would be different. However, we have already shown that in many cases the field tuning of position cannot be reconstructed by the speed/direction tuning (**Fig. 2e** for example). Therefore, the position tuning we observed cannot be attributed to causal signals that hold the muscle in a particular position.

Another evidence supporting that the tuning is more about the hand position than a specific causal signal controlling the muscle came from a previous study[3], which shows that microsimulation of different sites in the PMd caused monkeys to move the hand to particular positions. Importantly, as the initial positions of the hand were different when the stimulations were given, the muscle activity required to move the hand to that particular position differ significantly, suggesting that stimulations “did not specify a fixed set of muscle activations” [3] but a final hand position.

We have added a section in the discussion to better explain these issues.
(main text: line 175-189)

Comment 1 c): I notice even in the first and second examples, there is a hotspot of firing near the mouth of the animal. I was wondering whether some of the responses are due to peripersonal space/arm reaching space. There is also this notion of the arm reaching space as well in these brain areas. This leads to additional consequences for interpreting the results that the authors propose that this is a spatial encoding of hand position.

Reply 1 c):

It is relatively easy to exclude the interference of the arm reaching space (ARS), as the entire space we explored is, by definition, within the ARS. Regarding the peripersonal space (PPS), there were no tactile stimulations during our reaching tasks and the only visual stimulations within PPS were the fruit pieces. However, we have already excluded the interference from the reward location tunings in our identification criteria. Further, we note that the vast majority of the fields we observed were not near the mouth of the animal (see **Supplementary Fig. 6**). For a few fields that did very close to the mouth, we agree that future studies are needed to examine if the cell will respond when other objects rather than the hand are within the area, which will tell if it

is hand position tuning or PPS tuning. We have added related discussion in the revision.
(**main text: line 190-197**)

Comment 1 d): The assessment of the effect of reward needs to be quantified especially given that the supplementary figure currently just shows a few reward locations (“many did not correspond to the location where the reward was frequently given (Supplementary Fig. 8)”). Some statistical assessment of first regressing out reward location and then performing the place field analysis is the right way to do it. In fact the general approach I would take is to first regress the firing rate of the neuron as

FR = f(reward location, eye position, hand speed, hand direction) and then use the residuals and examine if they have place field encoding. Such an analysis is necessary in my opinion to claim the existence of a place field like tuning in the premotor cortex for hand position. I think such an analysis would assuage all the issues in major comment 1

Reply 1 d):

We fully agree with the reviewer that it is important to make sure that the hand position tuning we observed cannot be explained by other, known selectivity of the PMd neurons.

First, according to the suggestion, we regressed the firing rate of the neurons linearly using variables including the speed, direction of hand movements, as well as the location of reward:

$$\hat{r}_i = a_i * handspeed + b_i * handdirection + c_i * foodlocation_{onehot} + d_i$$

where \hat{r}_i is the regressed firing rate of neuron i and a_i , b_i , c_i are regression coefficients, d_i is the regression constant. The eye position was not included within the regressing because we did not record the eye movements during the tasks. We smoothed the raw firing rate with a 3-bin-width Gaussian window before estimating the regression coefficients and constant. To dissociate the effects of hand position and reward location, we excluded periods when the monkey grasped the food and returned backward to its mouth because the movement of the hand and the movement of the food were the same during this process. We made the one-hot encoding for food locations (the number of total possible spatial bins are $\sim 40 \times 40$).

Second, we calculated the regression residuals and used them to evaluate if there existed field-like tuning for the hand position. The calculation of new spatial “firing rate” maps, spatial mutual information and shuffled data were the same as in the manuscript (see **Methods: Spatial firing-rate map, Measurements used for cell type classification and Shuffling and classification criterion** for details).

After going through such a process, only 2 out of 132 apparent hand position cells in the manuscript (**Fig. 2a**, Hand Position-tuned cells) remained to be significantly position selective.

However, the regression process ensures that the information regarding the property of interest (i.e., the hand position) that can be attributed to other variables (i.e., speed, direction, and reward position) will be extracted from the source variable, without

properly delineating the actual contribution. As a result, it is too conservative a measure to judge the existence of any new tuning property. In other words, this procedure may produce too many false-negative judgments.

To make a rough estimation of to which degree the regression approach can produce false-negative results, we applied the same procedure to the population of 144 apparent direction tuning cells (**Fig. 2a**, Hand Direction-tuned cells), using the moving speed, hand position and reward position as the regressors and then calculating the residuals to identify the “direction” cells. Only 8 out 144 cells passed the test, although the moving direction tuning is a well-established property of the PMd neurons. Thus, we tend to conclude that this regression approach cannot give an accurate estimation regarding the strength and prevalence of hand position tuning in this case.

Then to measure and exclude the influence of reward position on our conclusion, we used the same reconstruction analysis for controlling the influence of moving speed/direction. The reconstruction analysis has been shown to be effective in properly judging if newly found feature selectivity is attributable to previously known selectivities[5], [6] (to justify the effectiveness of the reconstruction analysis, please refer to our **Reply 5 b) to Reviewer #3** for details). The advantages of reconstruction analysis include: 1) it eliminates interference by comparing the relative size of information from different variables; 2) it is a model-free process and therefore do not require any assumption regarding the data.

As we described in the original paper, after the reconstruction analysis considering the speed/direction tuning, there are 80 cells (out of 132 apparent hand position cells to begin with) exhibit hand position selectivity. Then these 80 cells were further subjected to the reconstruction analysis considering the reward position tuning. As a result, 50 out of 80 cells remained. This result indicates that the hand position tuning in at least some PMd neurons is indeed not dependent on known tuning for hand moving speed, direction, and reward position.

The reconstruction analysis of reward location was added to the revision, and all subsequent analyses regarding the properties of hand-position cells were recalculated based on these 50 cells, rather than the previous 80-cell population. Accordingly, **Fig. 1-3** and **Supplementary Fig. 6-13** were updated.
(main text: line 103-121)

Minor Comments:

Comment 2: I noticed some of the arrays are more in PMd and others are more in PMdr. Are there differences between the amount of hand position tuning in PMdr compared to PMd.

Reply 2:

We show the photos of the implanted electrode arrays in **Reply Fig. 4**. To answer this question, we compared the proportion of hand position cells across different array

locations. We find that Monkey Z and Monkey X (closer to PMdr) have a similar proportion of hand position cells (Monkey Z: 6.5%, 15/230; Monkey X: 6.8%, 17/248), indicating that there is no difference between the hand position tuning in PMdr compared to PMd. Monkey B (relatively closer to M1) has a greater proportion of hand position cells of 21.7% (18/83). As a previous study has reported microstimulation to both premotor and primary motor cortices can elicit arm movements towards specific target location[3], future studies need to explore in more posterior areas that adjacent to M1 or even M1 itself to see if there are more hand position cells there. No hand position cell was identified in Monkey A (0%, 0/40), probably due to the relatively small number of cells recorded from this array. (**main text: line 221-234**)

During the revision, we have tried to re-scan CT and MRI to determine the exact locations of implanted arrays relative to the cortical landscape (e.g., the central sulcus). Unfortunately, it was not successful due to the artifacts on the images caused by of the implanted titanium headposts and electrode pedestals.

We now provide more information regarding the hand position cells and the recording site in the revision, and discussed future work needed to be done to better elucidate this issue (**main text: line 221-246**).

Reply Fig. 4. Photos of implanted arrays in four monkeys. The arrays implanted in the prefrontal area close to the principal sulcus were not included in the present analyses and are masked in the photos. The second PMd array in Monkey B was not functioning and therefore is also masked.

Comment 3: Figure 1d is very different from figure 1c. I would like to see a heatmap of Figure 1c plotted in Figure 1d with minimal post processing. Why is the hotspot near the mouth for instance not visible in Fig. 1d

Reply 3:

Fig. 1c is the map of hand trajectories superimposed with spikes. However, this map cannot be directly used to measure space specificity of firing, because it does not control for the time spend in different locations. For example, if the hand stays in one location (e.g., near the mouth) for a prolong period, more spikes would be recorded even if the cell has not spatial preference whatsoever. To control the effects of differential time the monkeys spent in various locations, we calculated the spatial firing-rate map which measured the activity of cells in different locations as spikes/sec and **Fig. 1d** shows the resulting spatial firing-rate maps.

The calculation process from trajectories and spikes maps to spatial firing-rate maps is now described in detail in **Methods: Spatial firing-rate map**, which includes the following steps (see **Reply Figs. 5-7**):

- i) divide the moving space into small spatial bins, bin size: 1cm × 1cm (1st column).
- ii) count spikes in each spatial bin and get the spike count map, count time spent in each spatial bin and get the time map (2nd and 3rd column).
- iii) divide spike count map by time map and get the spatial firing rate map (4th column).

Reply Fig. 5. The calculation process of spatial rate map. We show the same cell in **Fig. 1 c & d**, marked as Cell 23 from Monkey Z in Session 7.

As we see in the figure, there are large number of spikes near the mouth, however the hand also spends long time near the mouth, so the hotspot near the mouth disappears on the firing rate map.

The calculation process of another three maps in **Fig. 1 c & d** are shown as below:

Monkey B Session 1 Cell 21

Monkey X Session 3 Cell 38

Monkey A Session 2 Cell 3

Reply Fig. 6. The calculation process of another three cells in Fig. 1 c & d.

In order to reduce the interference of noise on the estimation of spatial firing rates and for the purpose of visualization, we adopted a smoothing method[7]: the spike count map and time map are smoothed by a 2D Gaussian filter (standard deviation:

1.5 bin, Matlab function *imgaussfilt*), then we divide the smoothed spike count map by the smoothed time map to get the spatial firing rate maps shown in the manuscript. The smoothing does not change the position or relative strength of hotspot on the maps. Below we compare the unsmoothed and smoothed version of the maps to demonstrate it.

Monkey Z Session 7 Cell 23

Monkey B Session 1 Cell 21

Monkey X Session 3 Cell 38

Monkey A Session 2 Cell 3

Reply Fig. 7. The calculation of the smoothed maps of cells shown in **Fig. 1 c & d.**

In the revision, we have provided more detailed description of the processing steps leading to the firing rate maps and added a figure (**Supplementary Fig. 3**) illustrating the process using an exemplar cell.

We note that according to Reviewer's suggestion, we further took the reward location into consideration when searched for the hand position cells (see **Reply 1 for details**). As a result, we changed some exemplar cells illustrated in the original **Fig. 1 c & d.**

Reviewer #3:

The authors have presented a novel and intriguing finding of the activity and representations that can be found within the dorsal premotor cortex (DPC). In particular, using a simplified reaching task, they found that certain cells in the DPC encode the physical position of the hand, described as “position cells”. Further, the position cells that were identified were, in some cases, completely separate from representations of velocity and direction that have been reported previously. The size, location, and shape of the position fields associated with each position cell were characterized, and the presence of other spatial encoding cell types was tested. Summarily, while some other types of spatial encoding cells were found in minor quantities, the spatial map encoded in the activity of the DPC is entirely attributed to the position cells.

While these findings are very promising for advancing our understanding of the role of this premotor cortex, there are a variety of technical issues that must be resolved before considering this manuscript for acceptance.

Reply:

We are grateful for the reviewer’s thorough evaluation of the paper and constructive suggestions.

Major Changes:

Comment 1: In the current version of the manuscript, there appears to be a notable lack of technical control over the parameters of the task, as well as a lack of a reasonable and relevant control task. Please address the following questions in order to clarify the framework of the study.

- a)** The manuscript states that 100 trials are used per recording session. How many unique positions occur within the 100 trials?
- b)** What locations were maintained constant between different monkeys, or between different sessions for the same monkey?
- c)** How many trials are presented for each unique location?
- d)** Are all the positions maintained at exactly the same height (Z-dimension)?
- e)** If the locations are largely non-overlapping, how could unified conclusions be drawn from mismatched sets of locations?
- f)** Supplementary figure 8 and its caption suggests that there are few (2 or 3) positions for grasping. Such a characteristic does not agree with the mapping in the movement.
 - i)** Please be more detailed and add a supplementary figure with the spatial distribution of targets in the XY, XZ, and YZ planes to better elucidate the framework of the reaching task.
 - ii)** What purpose does isolating these few positions serve in developing the

manuscript's narrative?

Reply 1:

We are sorry for the lack of clarity in experiential details. Thanks for all the questions. Below we elaborate on more detailed experimental settings and our considerations. As a natural reach and grasp task, during the experiment, the experimenter arbitrarily chose the trajectories and end positions of the food pieces. Meanwhile, there was no explicit instruction or time limit for the monkeys to perform the task. Monkeys themselves initiated the reaching movement on their own pace, meaning that they could reach immediately when the food appeared, or wait until the food stopped before reaching out.

1 a): The arbitrarily chosen trajectories and end positions induce rich natural movements of the monkeys' upper limb in space. However, it prevents accurate counting on the exact number of different target positions.

1 b): As suggested, we now plot the trajectories of food (see **Reply Fig. 8**), which show both the 3D trajectories as well as their 2D boundaries. We can see that the areas maintained constant between different monkeys and overlapped across different sessions for the same monkey.

1 c): As explained in **a)**, unfortunately we cannot count the exact number of trials for each unique target location.

1 d): The locations of target were arbitrarily chosen in the 3D space, so the positions had different heights.

1 e): As we shown in **Reply Fig. 8**, the target locations are largely overlapped. What we proposed here is that some neurons are tuned for the hand position when the arm performing a reach and grasp task in front of the animals, which holds true regardless the how we choose the exact target locations, leading to a unified conclusion.

1 f-i): We plot the trajectories of food centered on monkey's nose in **Reply Fig. 8**.

1 f-ii): The original purpose of isolating these few positions was to exclude the interference of reward representation because the regions covered in our manuscript possibly have tuning properties of the reward.

Although the positions of food were arbitrarily chosen by the experimenter during the task, we found that the trajectories of monkey's hand were denser in several space positions in some experimental sessions and we suggest that there are two possible reasons: a) the experimenter had stereotypical behaviors resulting in a higher probability of the food appearing in these positions, which made the monkeys reached for these positions more often; b) the monkey's own grasping preference made it more inclined to grasp or intercept the food when the food was in particular positions, whether stayed still or passed through.

Both possibilities are reward-related, which gives us the opportunity to partially eliminate interference from the reward signal. If a cell is involved in encoding reward information, the hotspot of the cell should be the position where the food was grasped most frequently.

The original Supplementary Fig. 8 shows several counterexamples, that is, the

hotspots of the spatial firing rate map do not coincide with the reward high frequency position, which indicates that the position tuning of cells is not derived from the reward information.

Apparently, this analysis is only qualitative since the spatial distribution of targets is not limited to a few positions, as we show in **Reply Fig. 8**. What we want to emphasize in the original Supplementary Fig. 8 is not that there are only few grasping positions, but the monkeys grasp the food more frequently in some positions than in elsewhere, and the mismatch between the reward positions and hotspots positions suggests that the position-specific firings are not entirely derived from rewards.

Following the suggestion from **Comment 1 d) from Reviewer #2**, we now added more comprehensive analysis to assess the impact of reward signal, which provide quantitative evidence that the hand position tuning is not entirely derived from the position of rewards (**Fig. 2**). Thus, to make the paper more succinct, we removed previous Supplementary Fig. 8.

We added **Reply Fig. 8** as **Supplementary Fig. 1** to show the distribution of food position in the experiments.

Reply Fig. 8. The spatial trajectories of targets in the XY, XZ, and YZ planes. We plot the trajectories of reward with reference to the position of monkey's nose from different sessions into the same figure. Different colors indicate different sessions. The red dot indicates the monkey's nose. Both trajectories in 3D space and boundaries projected into three 2D planes are shown.

Comment 2: Beyond the parameters of the task itself, the authors have failed to specify how the video recordings were coordinated with activity patterns from the

recorded neurons. How was video synchronized to neural activity? This detail is critical due to the fact that most programs need a common event recorded in both signals to use as a reference.

Reply 2:

Thanks for the question. We now added more descriptions about this issue in the section **Methods: Synchronization between videos and neural recording** which will also be explained below.

The camera (Hikvision) we used supports hardware-trigger mode, that is, when the camera receives a rising edge voltage (TTL 0 to TTL 1), the camera takes one frame image. We used a signal generator (DG1022U, RIGOL Technologies) to generate the trigger signal of periodic square wave to control the frame rate of the recording camera. The same trigger signal was connected to four cameras, so we could record multiple videos from different angles simultaneously. In addition, the same trigger signal was also fed into the trigger-in/marker port of the neural recording system (CerePlex Direct, Blackrock Neurotech or Intan Recording Controller, Intan Inc.). Thus, whenever a rising edge was generated by the signal generator, four cameras each captured one frame of image while the neural recording system recorded the timestamp of this rising edge, which will be referred as the timestamp of the video frame along with raw neural data.

This information is now added to Methods.

Comment 2 a) Additionally, a time jitter will occur with such a great mismatch between sampling frequencies, which the authors should have accounted for, although this procedure has also been omitted from the manuscript.

Reply 2 a):

We apologize if we do not understand this question properly. The sampling frequency of the neural recording system, i.e., the time resolution of the triggering events is 20K Hz (Intan system) or 30K Hz (BlackRock system), which is far beyond the frequency of the triggering signal itself (60-100 Hz). Thus, we can accurately synchronize the neural data with corresponding hand movements extracted from the video recordings.

Comment 3: Related with the previous point. In Figure 1, the reach trajectory graphs (Fig. 1C) seem to lack some corresponding representation when compared against the spatial density plots (Fig. 1D). I have included 3 concrete examples that should be individually addressed:

a) Cell 43 (Fig. 1C, left) has the greatest density of spikes very close to the “nose” location of the west section. Further, there is clearly observable activity in the northeast section. Why are these representations missing from the corresponding spatial plane that was generated (Fig. 1D, left)?

b) Cell 21 responds strongly near the nose, both below and above it, with a density of spikes that appears to be the most prominent across all the reach trajectories. The

west and northwest sections of Fig. 1D (middle-left) are lacking any visible indication of these dense areas.

c) Cell 3 responds with the greatest density in the south section of the plane (Fig. 1C, right), although this is missing from the spatial map (Fig. 1D, right).

d) Please clarify why these results are consistent between different cells when the visualized trajectories do not coincide between the 4 visualized cells?

Reply 3:

We apologize for not clearly explain this procedure. Reviewer #2 has raised the same question. Please ref to our **Reply 3 to Reviewer #2** for the answers and how we improve the description in the revision.

Comment 4: While the authors have made an effort to characterize the heterogeneity in the recorded DPC population, their process for sub-selecting groups with relevant and similar dynamics could be biasing the observed results.

a) It is important to apply a population dynamics analysis to the general, trial-averaged activity of all the neurons, not just the “relevant” ones.

b) Further, the idea of a spatial map should be a property that is intrinsically associated to the individual neurons, as the authors have claimed in the present work. Based on this conclusion, it is necessary to apply an analysis of population dynamics based on the individual trials, which could be feasibly computed using the Latent Factor Analysis via Dynamical Systems (LFADS) metric. For examples of implementation of this metric, see Pandarinath et. al, 2018 (Nature Methods).

c) How do the dynamics of individual trials relate to the general population dynamics from trial-averaged activity?

d) The population dynamics should be useful in characterizing the spatial map in a more compact format, which would aid the interpretability of Fig. 1, 2, 3, although especially so for Fig. 1C

Reply 4 a)-d):

We appreciate the reviewer’s question regarding the relation between the spatial map we reported here at one side, and the approach of using population dynamics to study movement at the other side. We understand that our results are more related to the traditional “representation” point of view, which emphasizes the feature selectivity at the level of individual neuron, while the population dynamics approach focuses on the generation of neuronal activities and corresponding movement patterns from the perspective of dynamics. It would be very interesting and potentially useful to combine these two perspectives.

Unfortunately, we found that the above-mentioned goal is difficult to achieve in the current manuscript.

Most of widely used methods to compute the latent variables, such as LFADS mentioned by the reviewer, require the well-defined structure of “trial” and “condition”. A well-defined trial needs to be a specific temporal window contains clear-cut “event”,

such as onset of cues, starting of specific movement, etc. which is difficult to define in our monkey-self-paced natural reach and grasp task. Similarly, as the food position and arm movement trajectories are arbitrarily chosen and varied on a trial-by-trial basis, we cannot identify a group of trials as well-defined condition. Without such clear structured trials and conditions, it is difficult to apply the population dynamic analysis to our data.

However, we can use a simple model to demonstrate that the spatial fields tuning of neurons and population dynamics are not necessarily conflicting with each other. Instead, there may be intrinsic link between them that needs to be revealed in the future.

Specifically, in the model, we assume a population of neuron with pure hand position tuning, with the Gaussian position fields randomly spanned the reachable 2D space (**Reply Fig. 9a**). We simulated a simple center-out reaching task, by define the starting point of the hand at the center of the space, and four target position at a separate position to the north, west, south, and east of the center point (**Reply Fig. 9b**). A trial is started from the hand moving from the center point to one of the targets, along a straight line with minor random fluctuations added. For each condition (e.g., target) we carried out 100 trials.

In each trial, the simulation hand passes through a series of position fields of neurons, producing specific patterns of population activities as a function of time, i.e., population dynamics. Conceivably, the population dynamics are among trials moving towards the same target and are quite different for trials moving towards different targets. Interestingly, if we apply a simple jPCA analyses[8] to such dataset, we can obtain the low-dimensional visualization of population dynamics that share important features with that of previously published neuronal dynamics, including separation of trajectories in the space, rotational shape of the trajectories, etc. (**Reply Fig. 9c**)

Although the model and simulation are simplistic in nature, they suggest that the representation point of view presented by the spatial map we discovered, and the population dynamics can co-exist. To better understand the relation between the two would be desirable, but we hope the reviewer can agree with us that it is somewhat beyond the scope of the current paper. In the present manuscript, we would like to focus on reporting and validate the discovery of the spatial map itself and would leave the exploration of its relation to dynamics to later studies.

We have added a section in the discussion part of the revision (**main text: line 204-211**) regarding this matter.

Reply Fig. 9. The population dynamics of a simulated position cell group. We simulated 64 cells which have pure position tunings. Neuron activities were gaussian functions of the position with various standard variation from 2 to 16. The centers of the field were randomly chosen within a 50×50 two dimensional space. **(a)** the spatial firing rate maps of the 64 cells. **(b)** hand trajectories in a simple center out reaching task. In each trial, hand starts from the center of the space, reaching to one of the four targets. Different colors distinguish different reaching directions. We repeated 100 trials for each target. Trajectories in each is affected by minor random fluctuations. Combining the tuning maps of the space in **(a)** and moving trajectories in the space in **(b)**, we could easily simulate the firing activities for each neuron through a poisson process. **(c)** the first plane of the jPCA latent variables.

Comment 5: The decoding of varying physical variables related to the reaching task in Fig. 4 (**Fig. 2**) is not sufficiently justified with references, and it is difficult to follow the analysis as it is in the current state. Maintaining a focus on the individual units is a glaring oversight, since the authors have clearly identified both separate, and common, subpopulations that can be studied as independent entities. Given that the populations representing direction and velocity are separated from the population representing the position, how can the authors determine that the position-tuned cells are not integrating velocity and direction signals, since these would suffice to pinpoint the physical location?

Reply 5:

Sorry for not clearly describing the decoding of varying physical variables related to the reaching task in the main text. Indeed, the relevance of these variables, as well as its calculation and the identification of cell-type in this regard, has been explained in

the **Methods** section. Please see **Methods: Spatial mutual information** to identify reward location tuned cells, **Methods: Speed score** to identify hand speed tuned cells and **Methods: Mean vector length** to identify hand direction tuned cells.

Regarding the relation among different cell-subpopulation, the reviewer raised an interesting possibility that the position cell gets the spatial information by integrating velocity and direction signals. This is related to the question of mechanisms that give rise to the position information in the PMd. Besides the above-mentioned possibility, the hand position cell can also infer the physical location by visual or proprioceptive inputs. This issue would be interesting topic to address in future studies, and we have added a paragraph to discuss it. Please also see our **Reply 1 b) to Reviewer #2** or **main text: line 212-220** for related information.

Comment 5 a) Please apply a separate analysis of population dynamics to this particular section of the study, since this could provide far stronger evidence for the isolated functionality of the purportedly separate populations.

Reply 5 a):

As we explain in **Reply 4 to this reviewer**, it is difficult to apply population dynamics to the current dataset. We will note the reviewer's insightful suggestions and will try to implement them later.

Comment 5 b) It is also essential for the authors to justify the metric used for the reconstructions, preferably using previous examples from recent work.

Reply 5 b):

Thank you for the question.

The reconstruction analysis we adopted was introduced and validated in [5]. The authors discovered a subpopulation of hippocampal CA1 cells with angular tuning to the goal direction and cells that encoded the distance to the goal. To verify that the goal-related signal is genuine and not biased by the place tuning or head direction tuning, they proposed the reconstruction analysis. The authors have verified the performance of this method through a simulation analysis, which we quote below to provide a more detailed description of its validation:

*To validate the performance of the analysis, we simulated pure place-cells: We generated spikes in a Poisson process based on a 2D Gaussian spatial tuning (centered for different simulations at every other spatial bin in the room), using the real flight behavior of the bat (taking the behavior from all days where goal-direction cells were recorded: $n = 49$ days; yielding a total of $n = 5,625$ simulated pure place-cells). The place-field size and peak firing-rate were based on the averages of the real place-cells in our data. This simulation showed that despite a possible behavioral inhomogeneity, the reconstruction analysis successfully reveals the true underlying tuning of the cells – note that the simulated pure place-cells all had lower error for the reconstruction that assumed pure place-tuning (fig. S3E, see **Reply Fig. 10a**): all black dots are below the diagonal: $n = 5,625$ dots). Accordingly, all the simulated place cells*

had goal/place index < 1 in Fig. 11 (pink histogram) (see Reply Fig. 10b).

This method is applicable to our data, so we adopted it to exclude the effect of hand direction, hand speed and reward location.

Reply Fig. 10. a. Distribution of errors for all the simulated pure place-cells (black dots; simulated using the real behavior of the bat for all days and all possible locations in the room, $n = 5,625$ simulated cells). Note that all the simulated place-cells fall below the identity-line ('goal/place index' < 1; see Methods), meaning that for all the simulated pure place-cells the reconstruction yielded a lower error when assuming pure place-tuning – confirming that the underlying tuning of these neurons was the place-tuning: as indeed we created them. **b.** Distribution of the goal/place index for all goal-direction cells (blue; $n = 58$) and for simulated pure place cells (pink; $n = 5625$). Goal-direction tuning could not be explained by pure place tuning ($P < 2 \times 10^{-4}$ for all 58 goal-direction cells; vertical lines, 99th and 100th percentile for the simulated cells). Adapted from [5].

Comment 6: Recent work in the DPC has shown that this area integrates both somatosensory information, as well as information from the visual sensory modality. How can the authors know that the cellular activity is related to the physical position of the hand, and not a trace related to integration of visual information?

Reply 6:

Great question. As we explained in the **Reply 6 b) to this reviewer** and **Reply 1 b) to Reviewer #2**, this is more about the origin of the position information, which will be addressed in future studies.

Comment 6 a) Have the authors recorded data in any primary visual areas or somatosensory areas?

Reply 6 a):

No. To trace back the information about hand position, we will try to record such

signals in future experiment. Thanks for the suggestion.

Comment 6 b) Does some control exist that can address this question? If so, this control needs to be highlighted, and if not, the authors should describe what kind of control scheme could allow for these doubts to be addressed.

Reply 6 b):

To investigate the contribution of visual or somatosensory inputs in determining the hand position for the PMd, we could 1) conduct the same experiment in totally dark environment, with only the reward can be seen (e.g., by using a faint fluorescent marker), and 2) block the somatosensory input of the arm, e.g., by injecting local anesthesia to the upper arm, which will selectively disrupts the visual and somatosensory inputs, respectively. To examine how the spatial map changes under such conditions will reveal the mechanisms underlying the formation of the map. We have added a section in the discussion part of the revision (**main text: line 212-220**) regarding this matter.

Comment 7: The decoder that is used in Figure 2 (**Fig. 3**) needs to be explained in greater detail. The populations of position cells and all cells need to be stated explicitly, including mentioning how the decoder's performance may change based on the differences between the response properties of the two separate populations. How are these two populations related to the populations indicated in Fig. 4 (**Fig. 2**)?

Reply 7:

According to the suggestion, we stated the populations of different cells explicitly in our manuscript (**main text: line 137, 140, 142**).

To further examine how the population of position cells could represent the hand position across the reachable space, we used identified position cells to decode the hand trajectory (cell number $n = 50$, averaged 4.2 ± 2.9 across individual sessions, see Methods: Trajectory decoding analysis). Overall, the correlation coefficient (r) between the observed and decoded trajectories across all sessions in all animals is 0.54 ± 0.10 (**Fig. 3h**). We also evaluated the decoding performances using the group of all putative SUs ($r = 0.68 \pm 0.17$, cell number $n = 839$, averaged 59.2 ± 31.9 across individual sessions) or the group of cells with stable XY-firing rate maps ($r = 0.68 \pm 0.16$, cell number $n = 601$, average 42.8 ± 18.6 across individual sessions). Though lower than the performance using all cells ($P < 0.01$, paired t-test) or stable cells ($P < 0.01$, paired t-test), the decoding performance using the hand position cells was significantly greater than shuffled data (**Fig. 3h**, $r = 0.10 \pm 0.14$, $P < 0.001$, paired t-test, see Methods). Furthermore, considering the number of cells involved in the decoding process, the position cell group utilized only 10% of the neurons to achieve 84% of the decoding performance, compared to all cell group (10% of neurons and 82% of decoding performance when comparing to stable cell group), suggesting that the position cells can efficiently represent the hand position across the reachable space.

The population of position cells is the same as the population of position-tuned cells in **Fig. 2 a**, and the population of stable cells includes all subpopulations in **Fig. 2 a** as different subpopulations of cells tuned by position, direction, speed, or reward are selected by calculating different criteria for stable (XY plane) cells. The population of all cells includes all putative single units after spike sorting.

Comment 7 a) If all the cells can be used to decode the position, then why are the authors focused only on the position cells?

Reply 7 a):

To examine the relative contribution of different subpopulations in decoding the position, we carried out the analyses explained above.

For potential practical use, we note that the PMd is a commonly used signal source for motor brain-machine interfaces (BMIs)[9]–[11]. In general, the more neurons we utilized, the better performance BMIs will achieve[12]. However, our decoding analysis showed that the position cell group recruited much fewer cells to achieve reasonable high decoding performance (see **Fig. 3h** and **main text: line 147-151**), suggesting that identifying and locating more task-related cells might be a more efficient way, rather than increasing neuron number blindly.

We mentioned this in our discussion section. (**main text: line 247-251**)

Comment 7 b) This decoder should be applied to the various subpopulations that have been described (839 putative single-units, 609 single-units with stable XY-spatial firing rate maps, 132 cells meeting the desired criteria)

Reply 7 b):

According to the suggestion, we added the decoding performance of using all putative single units, which achieved the highest performance (**Reply Fig. 11**, correlation coefficient: 0.68 ± 0.17) and we added this to **Fig. 3h**.

Reply Fig. 11. Distribution of decoding performance in XY plane of using all putative

single units, cells with stable firing rate maps and position cells. n.s., not significant, ** $P < 0.01$, *** $P < 0.001$, paired t-test.

Comment 8: In Fig. 2B (**Fig. 3g**), the X-centered trajectories and the Y-centered trajectories have vastly different scales (X: -20cm - 20cm; Y: -100cm - 100cm). Based on this, how can the X-graph (Fig. 2B, Top) be compared to the Y-graph (Fig. 2B, Bottom). Furthermore, the observed Y-centered trajectory diverges in many instances from the decoded trajectory from both populations of cells, which could be indicating that the information related to the Y-centered position is being contaminated by slight variations in the Z-dimension.

Reply 8:

In **Fig. 3g**, the zero-centered trajectory is actually the speed along the Y axis, rather than the y-position. It is an error, and we apologize for it. We have changed the figure with the correct Y-centered trajectories which have similar scales as the X-centered trajectories. Thanks for helping us spot this error.

Comment 8 a) I would like to ask the authors to apply the same decoder to static periods of no movement, as a control to demonstrate the decoder's efficacy

Reply 8 a):

As the reviewer suggested, we used the decoder trained in motion periods to decode the trajectories in static periods (**Reply Fig. 12**), and results showed that the decoding performances would decrease ($P < 0.05$, paired t-test) but were significantly greater than the shuffled performances ($P < 0.01$, paired t-test), suggesting that the representation of the hand position by these cells is not exclusively reliant upon the presence of motion itself. We added this figure as **Supplementary Fig. 13** and discussed this result in **main text: line 256-265**.

To get the shuffled decoding performances, we first shifted the spike train by a random interval then used 70% of data to train the decoder and remaining 30% of data to test the performance. This process was repeated for 100 times for each recording session and the averaging performance was calculated as the final shuffled performance for this session. We added this content to the **Methods: Trajectory decoding analysis**.

We show results from the population of position cells in XY plane, the same results can be obtained from the population of all putative single cells or stable cells in XZ plane or YZ plane (not shown).

Reply Fig. 12. The decoding performances in static periods (yellow dots) are lower than those in non-static periods (red dots) using the same decoder but greater than shuffled performances (green dots). * $P < 0.05$, ** $P < 0.01$, *** $P < 0.001$, paired t-test.

Comment 8 b) I would also like to recommend calculating the same graph for the Z-centered position

Reply 8 b):

According to the suggestion, we have added the results of the Z-centered trajectories in **Fig. 3g**. The shown results were based on the XZ plane.

Comment 8 c) Can the authors identify some reason in particular that could explain the discrepancy between decoding X- vs. Y-centered positions?

Reply 8 c):

As we explained in **Reply 8 to this reviewer**, in the updated/correct figure, there were actually no differences among the scales in three axes. We further compared the decoding performances between two axes in the same plane and found no significant difference ($P > 0.05$, paired t-test). This conclusion holds in all three planes.

Comment 9: It is unclear how properties that are included about the position fields of the cells (Fig. 3) are related to decoding the position.

Reply 9:

Considering the reviewer's suggestion, we have tested the relationship between properties of the position fields and the decoding performance (**Reply Fig. 13**). We found no significant relationship between the field properties and the decoding performance using the position cells at the moment.

Reply Fig. 13. The relationship between properties of position fields (field area coverage, spatial mutual information/SI sum, spatial sparsity sum and spatial coherence sum) and decoding performance using position cells.

Comment 9 a) What significance does an ovular position field hold to the overall conclusion of the article?

Reply 9 a):

An ovular position field means the field representation is anisotropic along axes in each single position cell, which can tell neuronal activity represent spatial information with different resolution along different directions. Further study can investigate if such a design represents an optimal population coding strategy. We added a sentence to make this point clear. (main text: line 252-253)

Comment 9 b) The graphs for spatial sparsity and spatial coherence are so similar,

so why include both? What benefit comes from having both?

Reply 9 b):

The spatial sparsity and spatial coherence are two different measurements of the spatial firing rate map of position cells. We explained their meanings in section **Methods: Spatial sparsity and Spatial coherence**.

Spatial sparsity measures the fraction of the space in which the cell is active. Spatial coherence is an estimation of the local orderliness of the spatial firing distribution of the cell.

We believe that it is only a coincidence that the statistical distributions of spatial sparsity and spatial coherence are similar, after we further excluded more cells after evaluating interference from reward tuning, the new graphs for spatial sparsity and spatial coherence became different (**Fig. 3a & b**).

Comment 9 c) What is the relationship between position field size/shape and sparsity/coherence?

Reply 9 c):

According to the reviewer's suggestion, we have tested the relationship between position field size/shape and spatial sparsity/coherence (**Reply Fig. 14**). We calculated the eccentricity of the field to measure the field size (see section **Methods: Field detection**). We found:

- a) positive relation between field size and sparsity in the XY and XZ planes (XY: $P < 0.01$, XZ: $P < 0.05$).
- b) no significant relation between field size and coherence.
- c) no significant relation between field shape and sparsity.
- d) no significant relation between field shape and coherence.

We think the finding a) is an expected result because the larger the sparsity value, the larger the field area. We also tested the relationship between field area coverage and sparsity and the conclusion holds (XY: $P < 0.001$, XZ: $P < 0.05$, not shown).

Reply Fig. 14. The relationship between position field size/shape and spatial sparsity/coherence.

Comment 9 d): Which are the optimal properties to decode the position fields?

Reply 9 d):

We apologize if we did not understand the reviewer's question of "decoding" correctly.

If the reviewer meant the relationship between the properties and the decoding performance of hand position, as we concluded in **Reply 9 to this reviewer**, we found no significant relationship between the field properties and the decoding performance using the position cells at the moment.

If the reviewer meant which properties could affect the existence of the position fields. As mentioned in section **Methods: Field detection**, we defined the firing field as a set of contiguous regions whose: a) firing rates were above half of the peak firing rate of

the map; b) area was larger than 25 cm² and c) visited by the hand at least five times. So, the answer should be that the value of firing rate in the region and the area size of such region both matters.

Comment 10: The current manuscript has focused on analyzing the recorded populations of neurons in the DPC, although their treatment on the subject is severely lacking with respect to population dynamics and heterogeneity observed across the population.

a) Reflect on Kaufman et. al, 2014 (Nature Neuroscience); this article frames the activity in DPC based on a limited gating of preparatory activity that permits isolation of execution dynamics. It is important to discuss how the spatial maps could interact with preparatory activity.

b) Recent work has identified mixed selectivity and associative activity patterns in the DPC. Please comment on the relationship between the spatial map and the decision-making frameworks that have been presented (Elsayed et. al, 2016, Nature Communications; Vyas et. al, 2020, Neuron)

c) Other works have shown that the DPC (PMd) plays a significant role in other cognitive processes related to abstraction (Diaz-deLeon et. al, 2022, PNAS), execution of movement (Churchland et. al, 2012, Nature; Wang et. al, 2019, Nature Communications; Glaser et. al, 2018, Nature Communications), and maintenance of decision signals (Rossi-Pool et. al, 2017, Neuron; Ohbayashi et. al, 2003, Science)

d) How does a spatial map and representation interact with the different signals and heterogeneity that has been described in recent work?

Reply 10 a)-d):

We have re-written the section **Discussion** according to the reviewer's suggestion to discuss the relationship between the position tuning and many other properties or functions in DPC.

The simplified, natural reaching tasks we employed lacked explicit and distinguishable periods demanding for preparations or decisions. This leaves the questions of how position cells are involved into the broader functions because PMd plays a significant role in many motor cognitive processes including preparation and execution of movement[8], [13], [14], abstraction[15], decision signals[16], [17] and motor learning[18] etc. We derived our main findings based on the non-stationary period data (speed greater than 1 cm/s to avoid the effects of changes in moving status, **Methods: Hand Motion Tracking**), then we applied the same decoder trained in these data directly to static periods of no movement (see **Methods: Trajectory decoding analysis**) and found that the decoding performances decreased ($P < 0.05$, paired t-test) but were still significantly greater than the shuffled performances ($P < 0.01$, paired t-test) (see **Supplementary Fig. 13**), suggesting that the representation of the hand position by these cells is not exclusively reliant upon the presence of motion itself. Considering that the final hand position of an arm movement is an especially prominent parameter even during the preparatory period, it would be

interesting to study how the activity of position cells can contribute to the movement planning. Given the high efficiency of position field tuning encodings demonstrated in the present study and others [19], [20], it would be promising to explore whether PMd neurons adopt a similar field tuning framework to represent the 'decision space' or other non-spatial 'stimuli map'. (**main text: line 254-270**)

Comment 11: The activity observed in DPC can be related to the activity that has been previously described in the hippocampus. The authors mention this point briefly, although it should be discussed with more care and depth.

a) What traits of the activity in DPC are directly relatable to the traits of the activity in the hippocampus?

b) Are there properties of the activity observed in DPC that are completely distinct from those observed in the hippocampus? If so, what are they, and what does their presence indicate about the conclusions that the authors have reached?

c) Has the hippocampus been studied in this task? Has it been studied in any similar tasks? If not, how can the authors claim any association between the activity in DPC and hippocampus?

Reply 11:

We are sorry that our mention of hippocampus activity misled the reviewer. After observing the phenomenon of field-like activity in the firing rate map of DPC neurons, we immediately thought of the famous place cell in the hippocampus. We think that the similarity in field tuning form (in the spatial firing rate map) between DPC and hippocampus may indicate a universal spatial information encoding framework beyond brain regions, although this association is only conceptual at this point.

We cannot answer the questions **a)-c)** at the moment because we haven't done an in-depth comparison and analysis between DPC and hippocampus, we need to design appropriate behavioral tasks and record signals from both brain regions.

We have reduced the content of the hippocampus and only made restraint mentions in the section **Discussion**.

Comment 12: The discussion section of the current manuscript is lacking in its consideration of the novelties presented recently in the field, and it does not place the presented results in an adequate context for understanding how the observed results are related to the general body of work done in DPC, as well as the body of work dedicated to reaching tasks.

Comment 12 a) Reflect on the context of DPC and how the results can be considered part of, or contradicting, the previously considered models.

Reply 12 a):

We believed that the discovery of hand position cells has broadened our understanding of how PMd represents spatial information. Specifically, we confirmed

the well-documented selectivity for hand moving direction and speed[21]–[25] in natural reaching movements and demonstrated its co-existence with the field tunings of hand position, manifested as different selectivity in separate but close-by neurons or mixed selectivity in the same neuron. Intriguingly, despite its close association with movement direction and speed, our findings underscored the surprising necessity for a direct representation of hand position, even within such simple, spontaneous reaching tasks. More natural and complex motor contexts require high-dimensional representations with mixed selectivity of multiple task-related variables at the level of individual neurons as well as neuronal ensembles[26], which may account for the currently observed multi-facet tuning of PMd neurons. This finding requires us to pay more attention to the mixed selectivity of neurons under natural, complex behavior and explore the population activities from more other perspectives, such as neural dynamics.

We have added a discussion of this issue to the paper (**main text: line 198-211**).

Comment 12 b) It is essential to return to the consideration of place and grid cells, and how they relate to the putative “hand-place” cells that were observed in DPC

Reply 12 b):

As we mentioned in our **Reply 11 to this reviewer**, currently we do not know the relationship between place cell/grid cell and the hand position cells during the same task. We suggest that the apparent similarity between them may indicate a general spatial information encoding framework across the brain, from the limbic system to the neocortex. (**main text: line 271-276**)

Comment 12 c) Reflect on the relationship between population dynamics and the spatial map. Is the spatial map solely represented in “position” cells? Can spatial maps be decoded from other, place-ignorant cell subpopulations?

Reply 12 c):

We are sorry if we did not fully understand the reviewer’s question.

If the reviewer asked whether there are other neurons representing spatial information. Our answer is Yes. PMd neurons have been associated with the directions of reaching movements as we could utilize various methods such as population vector algorithm (PVA)[27] to decode the spatial information.

If the reviewer asked whether the tuning form of spatial firing rate map exists in other neurons. We suggest that the answer is No. Although, we always can calculate a spatial map for any recorded neurons as long as we recorded the motion data and spike activities. However, such spatial maps could just be random or byproducts of other tuning properties. To avoid these cases, we adopted three step procedures to identify significant and unbiased position cells. Therefore, we believe that only spatial maps from position cells are meaningful.

As for the relationship between population dynamics and the spatial map, we could not conduct such analysis because it is difficult to extract neural dynamics from our

current datasets for reasons which we explained in **Reply 4 to this reviewer**.

Comment 12 d) Consider mentioning Graziano, Taylor & Moore (2002, Neuron) (this point is not important at all)

Reply 12 d):

In this study, authors found that electrical microstimulation of the premotor cortex could evoke different, coordinated postures to form a map of hand positions around the body. More importantly, authors found that stimulations did not specify a fixed set of muscle activations but a final hand position. We are glad that the reviewer has drawn our attention to this study. We think it is highly relevant for our finding that there is a spatial map of hand position in the PMd. It is of great interests and importance to investigate if the map we reported here may explain their empirical findings. We are very grateful as the reviewer mentioned this work, and we have cited it and added related discussions to the revision. **(main text: line 183-189, 230-232)**

Comment 13: The Pearson correlation coefficients that are employed for calculating the speed score, the grid score, and the recording stability have no mention of the significance criterion used. This oversight is compounded by the lack of false positive validation. In the case of the recording stability the halved pieces of the dataset should be correlated with the full dataset, in order to have a reference frame for the correlation coefficient that is obtained.

Reply 13:

The description of the significance criterion was stated in section **Methods: Shuffling and classification criterion**:

To obtain the shuffled data, spike trains were shifted by a random interval that was more than 30 s and less than the session duration minus 30 s (wrapping the end circularly to the beginning). Then we re-calculated all measures mentioned above and repeated this procedure 500 times. If the raw measure was larger than the 99th percentile of the shuffled data, the cell was classified as the corresponding cell type. Measurement comparison was done within each session.

We use this criterion to compare spatial mutual information, border score, speed score and grid score.

In the case of recording stability, we reference the [7] and set 0.3 as the threshold.

According to your suggestion, we calculated correlations between the halved pieces of the dataset and the full dataset (see **Reply Fig. 15**), the results show that the majority of correlations are high, 97.7% of correlations are above 0.5. We will keep our original threshold because we only want to exclude some extremely unstable data.

Reply Fig. 15. The distribution of correlation coefficients between the halved pieces of dataset and the full dataset. Blue line indicates the threshold we used in the manuscript, with its value 0.3 marked left.

Minor Changes:

Comment 14: The introduction section is limited in length and scope, which greatly decreases the interpretability of the manuscript as a whole. Along with the corrections previously mentioned, please summarize the general results of the manuscript as the final paragraph of the introduction.

Reply 14:

Thank you for pointing out this problem in the manuscript. We added one summarizing paragraph to the section **Introduction**. (main text: line 40-50)

Comment 15: The legend for Fig. S2 (**Supplementary Fig. 4**) is entirely lacking, with no description of what is contained within the figure. Please focus on describing the differences between each of the panels, and the significance of including them for the resulting conclusions.

Reply 15:

We are sorry for the negligence, and we enriched the description of this figure.

Comment 16: Mutual information estimation usually needs a correction due to the finite sampling method, especially when considering the fact that the underlying distributions need to be well estimated. A variety of shuffling techniques have been proposed to address this particular problem, although the authors make no mention of such a procedure.

Reply 16:

We agree that the mutual information estimation should be considered carefully due to the finite samples. As we stated in section **Methods: Spatial mutual information**, we calculated the spatial mutual information (SI) using the formula:

$$I = \sum_i P_i \frac{\lambda_i}{\lambda} \log_2 \frac{\lambda_i}{\lambda}$$

where P_i denotes the occupancy probability in the i th bin (after we divided the moving space into nonoverlapping spatial bins, the spatial location can be represented by an integer-value random variable i), λ_i denotes the firing rate of the cell in the i th bin and λ denotes the overall mean firing rate of the cell. Such formula requires us to provide good estimates of the firing rate in the space.

To get better estimation of neurons' responses in different locations, we adopted a simple regularization method, that is, we smoothed the spike count map and time map using a Gaussian kernel respectively to get a smoothed spatial firing rate map (for detailed calculating process, please see **Reply 3 to Reviewer #2**). Furthermore, spatial bins with little time spent (< 0.05 s) were discarded from the analysis to prevent poor estimates from affecting subsequent calculations. After these processes, we did not perform a further correction.

But again, as the reviewer emphasized, it is important to ensure that the limited sampling did not inject large bias. To this end, we adopted a bootstrap procedure proposed in [28], follows from considering the case of uncorrelated pairs: it involves generating a shuffled probability distribution by randomly pairing stimuli and real responses (which refer to the hand position and firing rate in our experiment), calculating the shuffled information contained in the real response about the randomly paired pseudostimuli and finally subtracting a fraction of the shuffled information from the raw value of measured information. Let I is the raw value of measured information and I_b is the bootstrapped estimate of the value, and the unbiased estimate of the value should be calculated as follow:

$$\hat{I} = \left[1 - \left(\frac{I_b}{I} \right)^\gamma \right] \times I$$

where γ controls the conservative degree of the correction. We shuffled 500 times for each neuron during the bootstrap procedure and set $\gamma = 2$ following [28]. The results are shown in **Reply Fig. 16**.

First, as we can see from the figure, the distribution of the corrected estimations of spatial mutual information shows weak differences to our original estimations (**Reply Fig. 16 a**) and the corrections to the values are usually very small (**Reply Fig. 16 b**), which indicate that our simple but useful regularization method is effective enough to our current data.

Second, to clarify if neurons are space informative, we adopted a shuffling procedure to calculate the shuffled distribution of spatial information and we set a threshold of 99th of this distribution (for details, please see **Methods: Shuffling and classification criterion**). This rigorous threshold can largely avoid misidentifying position cells due to the overestimated information value[29].

Third, in practice, it is estimated that ~twenty minutes of data is adequate for

measuring position-dependence (place cell) in hippocampus using the mutual information[5], [30]–[33]. The average recording time in our experiments is 26.3 ± 6.5 minutes, plus higher discharging activities in the PMd than in the hippocampus.

Taken together, we think that the issue of finite sampling did not interfere with our results in any significant way.

Reply Fig. 16. (a) the distribution of our estimation of mi in the manuscript (blue) and the distribution of the corrected mi using a bootstrap procedure (red). (b) the distribution of the correction value.

Comment 16 a) How many trials were used to calculate the spatial information related to each individual location?

Reply 16 a):

In the process of calculating the spatial firing rate maps (shown in our **Reply 3 to Reviewer #2**), we used all “trials” to estimate the firing rate maps. For each individual location, it is hard to count the exact number of trajectories passing through. Instead, we could count the time spent in each location during the experiment and use this to measure the samplings for each location.

Comment 16 b) Please describe how distributions were estimated, and if/how these distributions varied by neuron, monkey, or session.

Reply 16 b):

To estimate the distribution of sampling frequencies, we divided the time count map by the all-time spent in the corresponding session to get the probability of the hand to be in each spatial bin. So, these distributions were affected by how the monkey moved its hand in each session and the distributions were the same among neurons from the same monkey within one session.

Comment 17: The cells that are used as examples in Fig. 1C & D are abandoned for

a 5th, unrelated cell in Fig. 1E & F. Further, this 5th cell is not from any of the recording sessions that were included previously. This discrepancy may be understood as suspect, and it could even give the impression that only the X and Y dimensions were measured for the first 4 neurons.

Reply 17:

We appreciate it very much for pointing out such mismatch, and we have selected one cell from **Fig. 1 c & d** to show in **Fig. 1 e & f** to eliminate any possible confusion.

Comment 18: The authors have purportedly shown that cells in the DPC maintain information about the location (“place”) of the hand in 3-dimensional space. They proceed to analyze whether these same cells have qualities related to grid or border cells, but the authors fail to mention what the relevance of identifying grid cells or border cells is to the primary finding of the manuscript. Without this additional information, the last section of results holds almost no bearing on the general conclusions that are reached.

Reply 18:

Our initial intention is to enrich the content about different types of location tuning. We agree with the reviewer that this section seems to dilute the focus of the manuscript and we moved it to **Supplementary Materials**.

Comment 19: The authors have focused their analyses on analyzing the firing rate from the relevant group of neurons, although they have omitted the parameter details used to calculate the firing rate as a function of time.

Comment 19 a) These details are essential to understanding the study as a whole, so they must be specified.

Reply 19 a):

Thanks for the question.

There are two analyses in the manuscript where the firing rate as a function of time is needed:

First, calculating speed score to identify the hand speed cells. As described in our **Reply 2 to this reviewer**, we have a signal generator producing a precise, periodic trigger signals in ≥ 60 Hz, which were fed into neural recordings devices as well as the recording cameras. Thus, to calculate the firing rates, we simply use the timestamps of two consecutive trigger to define the time window to calculate the firing rate as a function of time. We added this content to **Methods: Calculating motion data and firing rate** to better describe the details.

Second, hand trajectories decoding. We already described the parameter details of calculating the firing rate as a function of time in section **Methods: Trajectory decoding analysis**.

Other analyses we used such as spatial mutual information, spatial sparsity, spatial coherence and so on were conducted based on spatial firing rate map which was the firing rate as a function of space, we already described the parameters of calculating such spatial firing rate map in section **Methods: Spatial firing-rate map** and we have shown more details in our **Reply 3 to Reviewer #2**.

Comment 19 b) The activity rates of different neurons vary greatly. How can these different rates of activity be used to compare single-units? One common solution is to normalize the rates of activity using a transform (e.g. z-score transform), although this procedure is not mentioned anywhere in the manuscript.

Reply 19 b):

In the present study, we did not directly compare the firing rates, or other measurements (spatial mutual information, speed score, mean vector length, border score, grid score, etc.) among different neurons.

In calculating those measurements, the value of firing rates does not directly affect the magnitude of measurements. Neurons with higher firing rates don't necessarily have higher "scores". But the variety of firing rates in different locations within the firing rate map does. For example, if neuron A discharges evenly everywhere within the hand motion space and it has higher firing rate-20 Hz, while neuron B discharges only in the left part of the space and it has lower firing rate-1 Hz. The spike activity of neuron B will get higher spatial mutual information as it tells us more about hand position than neuron A. The example above is an ideal simple case, but it already shows that the absolute value of the firing rate is not important.

However, in some calculations, e.g., spatial mutual information, the normalization is indeed implemented. This can be seen from the **Eq. 2** in the **Methods: Spatial mutual information**. As this is a necessary part of their calculation, we don't emphasize it as a preprocessing.

Comment 20: In lines 136-137, the value is denoted as larger than the expected value by chance, so please include what the chance value should be. The same comment holds true for lines 139-142.

Reply 20:

As we stated in section **Methods: Shuffling and classification criterion**, we chose the 99th percentile of shuffled data as the comparing threshold, so the chance level is 0.01 by definition. We will mention this point explicitly in corresponding part. (**main text: line 91**)

Comment 21: The word "grid" is misspelled on line 285

Reply 21: Corrected. Thanks.

Reviewer #4:

In this manuscript, the authors examined whether cells in the dorsal premotor cortex (PMd) of macaque monkeys exhibit field tuning of hand position during a natural reach and-grasp task. To test the hypothesis, the monkeys were trained to reach and grasp fruit pieces from a wand held by the experimenter. While the monkey was performing the natural reach-and-grasp task, the authors recorded activity of PMd neurons and the 3D position of the right hand using 4 video cameras. Their analysis of activity during movements showed that a sizable proportion of PMd neurons exhibit significant spatial specificity of firing. A proportion of PMd neurons exhibited selectivity for hand moving direction (51%) and speed (24%). The authors concluded that their findings suggest that the PMd may utilize a similar encoding framework as the hippocampus to guide goal-directed limb movements.

Testing to see if the similar computational mechanisms as in the hippocampus exists in PMd is a unique attempt. The study analyzed data during the movements. In addition, the target fruit location was changed to a new position while the monkeys were reaching in some trials. These left whether the sustained activity before the movements also has similar spatial specificity as an unanswered question.

Reply:

We appreciate the reviewer's evaluation of the work and very helpful comments/suggestion.

Comment 1: In the abstract and introduction, the authors stated about the hippocampus (line 6, 12, 16, lines 19-31) more than the PMd (line 11, 15, lines 32-40) even though they did not do experiments in the hippocampus. Some readers may expect to see the comparison of data between the PMd and hippocampus. Please balance the proportion to reflect the actual experiments and results.

Reply 1:

We appreciate this good suggestion. We have revised the **Abstract** and **Introduction** accordingly to de-emphasize the link to the hippocampus, which made the paper more focused and easier to understand. (**main text: line 24-27**)

Comment 2: Locations of implanted array electrodes in Figure 1a makes an impression that some data may be recorded in M1 or PMdr. For example, in Monkey X, a part of the implanted array looks like in PMdr (rostral PMd). In monkey B, a part of the array looks like in M1. The results must be different between the PMdc, PMdr and M1. Would you add central sulcus to each map to clear this point?

Reply 2:

We agree with you that the locations of the electrodes should have been described more precisely. We are sorry we did not see any sign of the central sulcus within the skull window during the surgery. We show the raw photos of implanted arrays in **Reply Fig. 4**. During the revision, we have tried to re-scan CT and MRI to determine the exact locations of implanted arrays relative to the cortical landscape (e.g., the central sulcus). Unfortunately, it was not successful due to the artifacts on the images caused by of the implanted titanium headposts and electrode pedestals.

To answer this question, we compared the proportion of hand position cells across the monkeys instead. In short, there is no different between PMdc (6.5%, 15/230) and PMdr (6.8%, 17/248) and there is a greater proportion of hand position cells in M1 (18/83). As Reviewer #2 raised a similar question, please refer to our **Reply 2 to Reviewer #2** for more details. We also found significant differences between the properties of position cells from M1 and PMd (**Supplementary Fig. 12**).

We now provide more information regarding the hand position cells and the recording site in the revision, and discussed future work needed to be done to better elucidate this issue (**main text: line 221-246**).

Note: the proportion of hand position cells are calculated after we further eliminated the impact from reward locations according to **Comment 1 d) from Reviewer #2**.

Comment 3: In Figure 1c, what each red dot represents is not clearly written. Please write it in the method section.

Reply 3:

Thanks for the question. The red dot means that the cell discharged when the hand was in the corresponding position and the size of the red dot is proportional to the spike counts. We added the details in the legend of **Fig. 1** and the section **Methods: Calculating motion data and firing rate**.

Reference List:

- [1] A. P. Batista, G. Santhanam, B. M. Yu, S. I. Ryu, A. Afshar, and K. V. Shenoy, "Reference Frames for Reach Planning in Macaque Dorsal Premotor Cortex," *J. Neurophysiol.*, vol. 98, no. 2, pp. 966–983, Aug. 2007, doi: 10.1152/jn.00421.2006.
- [2] B. Pesaran, M. J. Nelson, and R. A. Andersen, "Dorsal Premotor Neurons Encode the Relative Position of the Hand, Eye, and Goal during Reach Planning," *Neuron*, vol. 51, no. 1, pp. 125–134, Jul. 2006, doi: 10.1016/j.neuron.2006.05.025.
- [3] M. S. . Graziano, C. S. . Taylor, and T. Moore, "Complex Movements Evoked by Microstimulation of Precentral Cortex," *Neuron*, vol. 34, no. 5, pp. 841–851,

- May 2002, doi: 10.1016/S0896-6273(02)00698-0.
- [4] W. Fang, J. Li, G. Qi, S. Li, M. Sigman, and L. Wang, “Statistical inference of body representation in the macaque brain,” *Proc. Natl. Acad. Sci.*, vol. 116, no. 40, pp. 20151–20157, Oct. 2019, doi: 10.1073/pnas.1902334116.
 - [5] A. Sarel, A. Finkelstein, L. Las, and N. Ulanovsky, “Vectorial representation of spatial goals in the hippocampus of bats,” *Science (80-.)*, vol. 355, no. 6321, pp. 176–180, Jan. 2017, doi: 10.1126/science.aak9589.
 - [6] A. Rubin, M. M. Yartsev, and N. Ulanovsky, “Encoding of head direction by hippocampal place cells in bats,” *J. Neurosci.*, vol. 34, no. 3, pp. 1067–1080, 2014, doi: 10.1523/JNEUROSCI.5393-12.2014.
 - [7] X. Long and S.-J. Zhang, “A novel somatosensory spatial navigation system outside the hippocampal formation,” *Cell Res.*, vol. 31, no. 6, pp. 649–663, Jun. 2021, doi: 10.1038/s41422-020-00448-8.
 - [8] M. M. Churchland *et al.*, “Neural population dynamics during reaching,” *Nature*, vol. 487, no. 7405, pp. 51–56, Jun. 2012, doi: 10.1038/nature11129.
 - [9] J. A. Gallego, M. G. Perich, R. H. Chowdhury, S. A. Solla, and L. E. Miller, “Long-term stability of cortical population dynamics underlying consistent behavior,” *Nat. Neurosci.*, vol. 23, no. 2, pp. 260–270, Feb. 2020, doi: 10.1038/s41593-019-0555-4.
 - [10] D. Peixoto *et al.*, “Decoding and perturbing decision states in real time,” *Nature*, vol. 591, no. 7851, pp. 604–609, Mar. 2021, doi: 10.1038/s41586-020-03181-9.
 - [11] F. R. Willett, D. T. Avansino, L. R. Hochberg, J. M. Henderson, and K. V. Shenoy, “High-performance brain-to-text communication via handwriting,” *Nature*, vol. 593, no. 7858, pp. 249–254, 2021, doi: 10.1038/s41586-021-03506-2.
 - [12] M. A. Lebedev *et al.*, “Future developments in brain-machine interface research,” *Clinics*, vol. 66, pp. 25–32, Jan. 2011, doi: 10.1590/S1807-59322011001300004.
 - [13] M. T. Kaufman, M. M. Churchland, S. I. Ryu, and K. V. Shenoy, “Cortical activity in the null space: permitting preparation without movement,” *Nat. Neurosci.*, vol. 17, no. 3, pp. 440–448, Mar. 2014, doi: 10.1038/nn.3643.
 - [14] G. F. Elsayed, A. H. Lara, M. T. Kaufman, M. M. Churchland, and J. P. Cunningham, “Reorganization between preparatory and movement population responses in motor cortex,” *Nat. Commun.*, vol. 7, no. 1, pp. 1–15, 2016, doi: 10.1038/ncomms13239.
 - [15] G. Diaz-deLeon *et al.*, “An abstract categorical decision code in dorsal premotor cortex,” *Proc. Natl. Acad. Sci.*, vol. 119, no. 50, Dec. 2022, doi: 10.1073/pnas.2214562119.
 - [16] M. Wang, C. Montanède, C. Chandrasekaran, D. Peixoto, K. V. Shenoy, and J. F. Kalaska, “Macaque dorsal premotor cortex exhibits decision-related activity only when specific stimulus–response associations are known,” *Nat. Commun.*, vol. 10, no. 1, p. 1793, Apr. 2019, doi: 10.1038/s41467-019-09460-y.

- [17] R. Rossi-Pool, A. Zainos, M. Alvarez, J. Zizumbo, J. Vergara, and R. Romo, "Decoding a Decision Process in the Neuronal Population of Dorsal Premotor Cortex," *Neuron*, vol. 96, no. 6, pp. 1432-1446.e7, Dec. 2017, doi: 10.1016/j.neuron.2017.11.023.
- [18] S. Vyas, D. J. O'Shea, S. I. Ryu, and K. V. Shenoy, "Causal Role of Motor Preparation during Error-Driven Learning," *Neuron*, vol. 106, no. 2, pp. 329-339.e4, Apr. 2020, doi: 10.1016/j.neuron.2020.01.019.
- [19] C. F. Doeller, C. Barry, and N. Burgess, "Evidence for grid cells in a human memory network," *Nature*, vol. 463, no. 7281, pp. 657-661, Feb. 2010, doi: 10.1038/nature08704.
- [20] A. Yin, P. H. Tseng, S. Rajangam, M. A. Lebedev, and M. A. L. Nicolelis, "Place Cell-Like Activity in the Primary Sensorimotor and Premotor Cortex During Monkey Whole-Body Navigation," *Sci. Rep.*, vol. 8, no. 1, p. 9184, Jun. 2018, doi: 10.1038/s41598-018-27472-4.
- [21] D. W. Moran and A. B. Schwartz, "Motor Cortical Representation of Speed and Direction During Reaching," *J. Neurophysiol.*, vol. 82, no. 5, pp. 2676-2692, Nov. 1999, doi: 10.1152/jn.1999.82.5.2676.
- [22] M. M. Churchland, G. Santhanam, and K. V. Shenoy, "Preparatory Activity in Premotor and Motor Cortex Reflects the Speed of the Upcoming Reach," *J. Neurophysiol.*, vol. 96, no. 6, pp. 3130-3146, Dec. 2006, doi: 10.1152/jn.00307.2006.
- [23] P. Cisek and J. F. Kalaska, "Simultaneous Encoding of Multiple Potential Reach Directions in Dorsal Premotor Cortex," *J. Neurophysiol.*, vol. 87, no. 2, pp. 1149-1154, Feb. 2002, doi: 10.1152/jn.00443.2001.
- [24] R. Caminiti, P. Johnson, C. Galli, S. Ferraina, and Y. Burnod, "Making arm movements within different parts of space: the premotor and motor cortical representation of a coordinate system for reaching to visual targets," *J. Neurosci.*, vol. 11, no. 5, pp. 1182-1197, May 1991, doi: 10.1523/JNEUROSCI.11-05-01182.1991.
- [25] Q. G. Fu, J. I. Suarez, and T. J. Ebner, "Neuronal specification of direction and distance during reaching movements in the superior precentral premotor area and primary motor cortex of monkeys," *J. Neurophysiol.*, vol. 70, no. 5, pp. 2097-2116, Nov. 1993, doi: 10.1152/jn.1993.70.5.2097.
- [26] S. Fusi, E. K. Miller, and M. Rigotti, "Why neurons mix: high dimensionality for higher cognition," *Curr. Opin. Neurobiol.*, vol. 37, pp. 66-74, Apr. 2016, doi: 10.1016/j.conb.2016.01.010.
- [27] A. P. Georgopoulos, A. B. Schwartz, and R. E. Kettner, "Neuronal population coding of movement direction," *Science (80-.)*, vol. 233, no. 4771, pp. 1416-1419, 1986, doi: 10.1126/SCIENCE.3749885.
- [28] L. M. Optican, T. J. Gawne, B. J. Richmond, and P. J. Joseph, "Unbiased measures of transmitted information and channel capacity from multivariate neuronal data," *Biol. Cybern.*, vol. 65, no. 5, pp. 305-310, Sep. 1991, doi: 10.1007/BF00216963.
- [29] W. E. Skaggs, B. L. McNaughton, M. A. Wilson, and C. A. Barnes, "Theta

- phase precession in hippocampal neuronal populations and the compression of temporal sequences,” *Hippocampus*, vol. 6, no. 2, pp. 149–172, 1996, doi: 10.1002/(SICI)1098-1063(1996)6:2<149::AID-HIPO6>3.0.CO;2-K.
- [30] W. Skaggs, B. McNaughton, and K. Gothard, “An Information theoretic approach to deciphering the Hippocampal code,” *Adv. Neural Inf. Process. Syst.*, vol. 5, 1992.
- [31] D. Mao, E. Avila, B. Caziot, J. Laurens, J. D. Dickman, and D. E. Angelaki, “Spatial modulation of hippocampal activity in freely moving macaques,” *Neuron*, vol. 109, no. 21, pp. 3521-3534.e6, Nov. 2021, doi: 10.1016/j.neuron.2021.09.032.
- [32] G. Ginosar, J. Aljadeff, Y. Burak, H. Sompolinsky, L. Las, and N. Ulanovsky, “Locally ordered representation of 3D space in the entorhinal cortex,” *Nature*, vol. 596, no. 7872, pp. 404–409, Aug. 2021, doi: 10.1038/s41586-021-03783-x.
- [33] R. M. Grieves, S. Jedidi-Ayoub, K. Mishchanchuk, A. Liu, S. Renaudineau, and K. J. Jeffery, “The place-cell representation of volumetric space in rats,” *Nat. Commun.*, vol. 11, no. 1, p. 789, Feb. 2020, doi: 10.1038/s41467-020-14611-7.

Reviewer #1

Overall, I think the authors did a nice job in re-writing the whole discussion and in addressing some of my points. As I stated for the previous version, I think the manuscript is well-written, but honestly, I still do not see how this study would advance what we know about the premotor/motor spatial encoding of movements.

For example, among important and relevant works, early studies showed that M1 neurons encode spatial gradients when holding the hand at specific orientations in a 2-D space (Georgopoulos et al., 1984, *Experimental Brain Research* (54)446–454) as well as in a 3D-space (Kettner et al., 1988, *J Neurosci* (8)2938-47).

These studies were later followed by the two Caminiti's papers (Caminiti et al., 1990, *J Neurosci*, (10)2039-2058; Caminiti et al., 1991, *J Neurosci*, (5)1182-1197) in which the authors clearly demonstrated that both premotor and motor cortices, beyond their coding of movement direction, have an internal representation of space for the arm movement (it is interesting to note that they did not use the typical center-out paradigm).

I feel the present study is a more “naturalistic version” (which is of course welcome and needed) of old studies, but it does not advance our knowledge of premotor/motor areas spatial encoding of arm movements.

Reply:

We appreciate the reviewer for highlighting these studies, which provide an opportunity to better demonstrate the novelty of our work. In brief, our novel finding is the discovery that the spatial coding in the PMd is in the form of “position fields”. Below we explain how it is different from previous studies and why it is significant.

At the factual level, we conducted further analyses to show that the gradient coding identified in early studies of M1 neurons (Georgopoulos et al., 1984; Kettner et al., 1988) do not account for the spatial selectivity observed in PMd neurons recorded in our study. Specifically, we compared the effectiveness of a 2D Gaussian function (representing spatial field coding) and a planar function (representing spatial gradient coding) in characterizing the current data. We found that the firing activities of 83% (109 of 132) of the hand position-tuned cells were well described by the Gaussian function ($R^2 = 0.80 \pm 0.19$, $P < 0.001$, F-test), while the planar function described only 50% (66 of 132) of these cells ($R^2 = 0.52 \pm 0.24$, $P < 0.001$, F-test). Moreover, the coefficient of determination (R^2) for the planar function was significantly lower than that for the Gaussian function ($P < 1e-25$, paired t-test), indicating that the field tuning hypothesis is significantly more robust than the hypothesis of gradient tuning. Importantly, this conclusion holds true whether considering the entire reachable space or just a part of it (the hotspot area in the map). These results strongly suggest that the gradient coding reported for M1 does not apply to the PMd.

At the conceptual level, we revised the text to clarify why the discovery of field spatial coding is a critical piece of information that reveals previously unknown aspects of the computations underlying motion planning. This also emphasizes the advancements of our current study compared to the work of Caminiti et al. (1990,

1991), which established that spatial information is encoded in M1 and the PMd but did not characterize the specific form of the tuning.

Revealing the specific tuning function of neurons is crucial for understanding the underlying neurocomputation, with notable examples including the concept of "receptive fields" for visual processing and "place cells" for spatial navigation. In our case, the field tuning of position in PMd establishes a vital connection to the well-established computational framework of the Tolman-Eichenbaum Machine (TEM) (Whittington et al., 2020, *Cell*), which requires positional field tuning. TEM represents the mainstream understanding of the computations underlying body navigation. Our findings suggest an intriguing link between TEM and limb movement, potentially inspiring new research avenues.

Through revision we were able to demonstrate that our findings are both factually new and conceptually significant. We are very grateful to the reviewer for helping us enhance the rigor and clarity of our work.

Reviewer #2:

Comment 1:

I apologize for the delay in returning the manuscript. The start of fall with attendant challenges made me delay my review. The authors have tried to address the comments of the reviewers with many different analyses. So that is a positive.

However, I unfortunately must say that I still remain unconvinced by some of the key analyses. To claim that there is a unique hand position tuned population of cells in PMd like the hippocampus needs a level of evidence that unfortunately is not present in the manuscript at this point. Based on the data provided, I cannot in good conscience recommend this paper for publication. I am not sure the 50 they find are uniquely hand position tuned because of the following problems. Note, this is one thing I dug into because I feel this was the one case where they performed analyses to address my comments, and it was unsatisfactory.

The point is ideally illustrated by the regression analysis. I might be taking an extreme position, but I fear “tuning” in the premotor cortex might be dead on conception. Basically, their analyses themselves reveal that because of the correlation between hand-speed, hand-direction, reward location, and eye position, they all are explanatory of the data. So, when you control for one variable, the others explain very little variance suggesting that the representation perhaps is not as simple as one thinks. I don’t even fully believe that there are unique hand direction cells in older experiments, they are just one of the many variables that experimenters tested. So, if you really want to make these claims, you need to experimentally decouple some of them. There are researchers who have done brutally hard experiments, where they make monkeys fixate in one location while reaching to the other to decouple eye centered from hand centered reference frames or reach in 3d using Wheatstone bridges and 100s of trials and even in those cases, the results are super complicated. The experiments in the paper lack such controls and thus the results are muddy.

Reply 1:

We have mulled over these pertinent comments. At the end, we fully agreed with you and realized that our data (and many other data if we looked closely, see below) did not support that these are unique cell populations dedicated for encoding a single variable. Actually, even the place cells in the hippocampus also represent other variables such as head direction and speed (e.g., Mao, D. et al., 2021, *Neuron*). A more accurate way to characterize the data is that the PMd neurons encode these task-related variables in a mixed coding framework. That is, for the majority of cells, they do not represent a single task-related variable, instead, several variables are simultaneously represented, with the exact way of mixture varying from cell to cell. Functionally, the mixed coding framework has been demonstrated to be beneficial for flexible linear decoding for complex behavioral tasks (Tye et al., 2024, *Neuron*; Fusi et al., 2016, *Current Opinion in Neurobiology*) and efficient and reliable neurocomputing (Johnston et al., 2020, *PLOS Computational Biology*). In the case of the PMd, it allows the spatial information, e.g., the hand position, target position, and

the kinetic information, e.g., the moving speed and moving direction, to be represented and integrated in the same population.

The claim that hand position is a dimension represented in such a mixed coding framework can be supported by the reconstruction analyses showing that in head-to-head comparisons with other parameters in terms of explaining data, hand position revealed itself as a non-negligible variable (please see the next reply for details).

We have made thorough revision to the manuscript to reflect this major change from claiming unique “hand position cells” to revealing that hand position is represented by neurons in a mixed coding framework, and such information is coded in the form of position fields. All related parts, including the title, abstract, introduction, results and discussion have been rewritten accordingly. We are very grateful for your insightful comments, which have helped us correctly interpret our own results.

Comment 2:

The authors perhaps might argue that I am being too conservative in my thinking and should go purely by the reconstruction analysis proposed. Even if one sets aside the regression analysis there are huge problems with the claims. Unlike the bat data, which is highly skewed outside the unity line, in the PMd data, the distribution looks symmetric around the unity line (compare Fig. 3c, d and the figure in the responses to the reviewers Fig. 10a). So, the authors basically take stuff outside the unity line from Fig. 3d in the position vs. reward reconstruction analysis. But when they look at the data using reward location basically everything falls on the unity line with some symmetric spread around it. In this case, would you not need to correct for multiple comparisons here and also assign a p-value for each cell based on a shuffle test. Doing so might reduce the number of “place cells” in the population even more. So overall even this analysis is unconvincing. And given that there are no eye movement controls, the 50 cells become even more suspect in my opinion. I pasted the figures at the bottom and the problem is apparent!

Reply 2:

We apologize that there is a misunderstanding regarding the figure we redrew from Sarel et al., 2017, *Science* showing the reconstruction analysis. The figure you mentioned that exhibits highly skewed data distribution outside the unity line was based on simulated “pure” place cells, not real bat data, which was aimed to validate the analysis in a “clean” situation. If we examine the real bat data, we can also observe that there are many points close to the diagonal.

We agree that assigning the datapoint to one of the two classes simply based on whether it is below or above the diagonal is not warranted. To be able to claim this in a convincing way, one needs to demonstrate that the deviation from the diagonal is statistically significant. However, as we no longer claim the existence of unique “position cells”, the reconstruction analysis is not used as a demonstration of dominance of one variable over the other. Instead, we refer to these analyses to show that in such head-to-head comparisons, the selectivity for hand position cannot be

ignored, as if it is the case, the distribution would be highly skewed towards the other variable. These analyses demonstrate that hand position is, indeed, a non-negligible component in the mixed-coding framework in the PMd. We have revised the results section to better explain this.

Reply Fig. 1. a. Reconstruction analysis of simulated data. Distribution of errors for all the simulated pure place-cells (black dots; simulated using the real behavior of the bat for all days and all possible locations in the room, $n = 5,625$ simulated cells). Note that all the simulated place-cells fall below the identity-line ('goal/place index' < 1), meaning that for all the simulated pure place-cells the reconstruction yielded a lower error when assuming pure place-tuning – confirming that the underlying tuning of these neurons was the place-tuning: as indeed we created them. Adapted from fig S3E. **b.** Population scatter for all the goal-direction cells ($n = 58$ neurons), showing the error values in the reconstruction analysis – comparing between the goal-direction signal and the head-direction signal for each neuron (dots). Note that 71% of the cells (41 of 58) were above the identity line, indicating that the cells' firing was explained better by the goal-direction tuning than by the head-direction tuning. Adapted from fig S5C.

Below we provide the Supplementary Materials for Sarel et al., 2017, *Science*, Page 12-13, fig. S3E, fig. S5C for your information:

To validate the performance of the analysis, we simulated pure place-cells: We generated spikes in a Poisson process based on a 2D Gaussian spatial tuning (centered for different simulations at every other spatial bin in the room), using the real flight behavior of the bat (taking the behavior from all days where goal-direction cells were recorded: $n = 49$ days; yielding a total of $n = 5,625$ simulated pure place-cells). The place-field size and peak firing-rate were based on the averages of the real place-cells in our data. This simulation showed that despite a possible behavioral inhomogeneity, the reconstruction analysis successfully reveals the true underlying tuning of the cells – note that the simulated pure place-cells all had lower error for the reconstruction that assumed pure place-tuning (fig. S3E): all black dots are below the diagonal: $n = 5,625$ dots).

Note that this reconstruction analysis has several advantages: First, it explicitly addresses the behavioral coupling between the position and the goal-direction. Second, it conserves the number of spikes. Third, the reconstruction analysis is model-free: the empirically-measured tuning curves serve as the model, and therefore do not require any model-fitting.

Comment 3:

In addition, given that these are fixed Utah arrays pooled across days, the number of units is also not unique. Maybe the same units are multiple counted on different days, which would reduce from 50 to even less. So after all these corrections, if we end up with say 3% of hand position tuned cells, is this biologically meaningful. In the bat data: All 58 of their cells were goal tuned and NOT place tuned.

Reply 3:

Now we no longer claim the “hand position cells”, thus the number of them would no longer be a concern. However, we value the question of potential multiple counting and conducted further analyses to investigate it.

To our knowledge, there is no widely adopted, standardized method for tracking single unit activity in Utah array for long time. To examine this issue, we used an algorithm from Fraser et. al., 2012, *Journal of Neurophysiology* to identify unique units recorded across multiple sessions, in which, pairwise cross-correlogram, autocorrelogram, waveform shape, and mean firing rate are integrated as identifying features of a neuron. The rationale behind this method is “*When two units recorded on separate days were compared using these features, their similarity scores tended to be either high, indicating two recordings from the same neuron, or low, indicating different neurons.*”

Out of the 839 putative single units extracted in our experiments, this method identified 451 (Monkey A, 30; Monkey B, 54; Monkey X, 238; Monkey Z, 129) “unique” neurons. We showed the lifetime for each unique neuron in the figure below (Reply Fig. 2). Furthermore, 53.4% (241/451) of these unique neurons were recorded on only one session and 46.6% (210/451) were recorded repeatedly over multiple sessions. We reported the details for each monkey in the table below (Reply Table 1).

As recounting would not only increase the number of cells showing position tuning, but also the total number of cells, its effect on the proportion is uncertain. Out of the 451 unique cells, we selected 340 cells with stable firing rate maps and identified 40 (11.8%) cells that were hand position-tuned following the same process in the main text. This number is actually slightly higher than the one we originally reported 8% (50/601), which did not take possible recounting into account. This analysis suggests that the potential issue of recounting would not change our conclusion in any significant way.

Considering that identifying unique cells recorded with Utah array is not a widely adopted step, we added these results to the supplementary material.

Reply Fig. 2. Lifetime for each identified unique single unit across multiple recording sessions. A black line segment indicates that the unit was recorded on a given session.

Reply Table 1. Number of repeatedly recorded sessions for each unique neurons in each monkey.

Subject \ Repeated Sessions	Repeated Sessions							
	1	2	3	4	5	6	7	8
Monkey A	15	9	6	-	-	-	-	-
Monkey B	22	18	13	1	-	-	-	-
Monkey X	143	59	20	13	1	2	-	-
Monkey Z	61	17	26	14	2	7	0	2
summary	241	103	65	28	3	9	0	2

Comment 4:

Perhaps, the authors want to say, see how good our decoding analyses are from these 50 cells. However, this has problems as well. Did the authors compare the decoding correlation for a random 50 cells from their population and show that the resulting correlations are different? A shuffle control is meaningless. Take a random 50 cells from the stable cells and repeat and show that the distributions are fully different. Such an analysis is necessary for the claims in the manuscript. They might

have. But the way the authors have written the paper with so many supplements and just 3 main figures that are sparsely described in figure legends makes it all impossible to follow.

Reply 4:

A very good suggestion indeed! Accordingly, we compared the decoding correlation to randomly selected cell groups rather than the shuffled control. The results confirmed that decoding superiority of the 50 cells showing position tuning ($P < 0.001$, paired t-test).

The method as well as the results were revised to reflect this change. Please see Methods: Trajectory decoding analysis-*Chance level*, and Main text, Figure 3H.

Reply Fig. 3. Distributions of decoding qualities (Pearson correlation) using different groups of cells and the distribution of the decoding chance performances. The group of all cells--all putative SUs, cell number $n = 839$, averaged 59.2 ± 31.9 across individual sessions; the group of stable cells--cell number $n = 601$, averaged 42.8 ± 18.6 across individual sessions; the group of significant/primary hand position-tuned cells (abbreviated as “position cell” in the figure) after reconstruction analysis, cell number $n = 50$, averaged 4.2 ± 2.9 across individual sessions. n.s., not significant, ** $P < 0.01$, *** $P < 0.001$, paired t-test. (Figure 3H in the revised paper).

Comment 5:

Overall, I am unsure this manuscript clears the bar for the claims in the manuscript. Let aside questions of novelty, impact etc. I just cannot be sure the experimental control and statistics support the conclusions in the manuscript. There may indeed be hand position tuning in PMd, but this dataset and analyses do not meet that bar. I do apologize if the tone in my review feels strident. But I wanted to voice my worries about the data in as clear a manner as possible and note everything cannot be a discussion point that will be addressed in future work!

Reply 5:

We appreciate your insightful and pertinent comments that guide us to refine the work. We demonstrate that the position tuning is a non-negligible component in a mixed coding framework in the PMd. We hope it will serve as a steppingstone to stimulate future investigations to examine this important matter.

REVIEWERS' COMMENTS

Reviewer #1 (Remarks to the Author):

This manuscript has undergone significant changes since the first submission. I think the authors have done a good job reframing the manuscript to be more suitable and coherent, and they have addressed my previous concerns. Upon re-reading the latest version, I have only two points that I think should be revised or clarified:

Comment 1) Perhaps I missed this in the previous versions, but the authors state, “we collected 839 (Monkey A, 51; Monkey B, 101; Monkey X, 390; and Monkey Z, 297) putative single units (SUs).” However, the analysis was conducted on 601 neurons. The selection process leading to this reduction is unclear.

Response 1:

Thanks for the question. To ensure the validity of our analysis of hand position tuning, we exclusively evaluated stably recorded neurons, thereby excluding those with unstable firing rate maps across single recording session from our analysis. The stability of spatial firing rate maps was assessed by dividing all recorded data within each session into two equal halves and computing two new firing rate maps accordingly. The Pearson correlation coefficient (r) between these two maps was then calculated. In line with the method in Long et al., *Cell Research*, 2021, we established a threshold of $r = 0.3$ to exclude neurons exhibiting extreme instability. Following this rigorous selection process, the total number of stable neurons included in the following analyses was 601.

This was explained in the original manuscript. Now we have highlighted this selection process in the revision (**main text, line 105-109**). Comprehensive details regarding this process are delineated in the **Methods: Recording stability** and **Supplementary Fig. 4**.

Comment 2) In the discussion, it is not clear how the authors conclude, “we argue that the field-like hand position coding in the PMd is egocentric and would remain constant after body relocation, which is distinct from the allocentric space representation for body navigation.”

Response 2:

We have reorganized the text to clarify the logic of our hypothesis of the stability of the field-like hand position coding in the PMd in the egocentric frame (**main text, line 238-242**):

Critically, as monkeys can execute consistent reach-and-grasp movements across varying body locations and orientations, we hypothesize that the field-like coding of hand position in the PMd provides robust hand positional encoding in the egocentric frame¹, thereby establishing a computational foundation for stable body-to-limb

coordinate transformations.

Reviewer #2 (Remarks to the Author):

I like the revisions and the reframing of the paper and it feels less dramatic as a claim. I think given the 4 monkeys worth of data and a major softening of the claims of the paper make it more suitable for publication now. I also like the natural behavior.

However a couple of other recommendations just to tighten the results and this could be reviewed by the editor and included in supp. information.

Comment 1) The authors should perform a neuron dropping analysis to take all the hand position tuned neurons and remove them from the decoding analysis in Fig. 3H for the all cells and stable cell population. If the decoding accuracy drops sharply that supports their claims even more. I mean even a random set is quite good, the performance is reasonable. To strengthen their claim, show the random reconstruction in a separate panel in Figure 3G perhaps with the position cells to bolster this claim.

Response 1:

As suggested, we removed all the hand position tuned cells from the all-cells population and the stable-cells population and found that the decoding performances indeed dropped further compared to the random selection condition (correlation coefficient = 0.31 ± 0.18 when removing from all cells, random-select cells1; and 0.33 ± 0.18 when removing from stable cells, random-select cells2). Both random-select cell groups demonstrated significantly lower decoding performance compared to the primary hand position-tuned cells ($P = 1.89 \times 10^{-4}$ and $P = 3.37 \times 10^{-4}$, $n = 18$ sessions, one-sided paired t-test). We have added these random-select performances in **Fig. 3H** and described the results in **main text line 197-205**. In addition, to ensure simplicity and easy understanding of the figures, the decoded trajectories of random-select cells are shown in **Supplementary Fig. 8** to highlight the comparison, which are not directly added to Fig. 3G. We thank the reviewer for this suggestion, which indeed further strengthened our claim.

Response Figure 1 (Supplementary Fig. 8). Examples of observed and decoded hand trajectories using random-select cells and primary hand position-tuned cells. Gray lines are the observed trajectories, blue lines are the decoded trajectories using random-select cells1, yellow lines are the decoded trajectories using random-select cells2, and red lines are the decoded trajectories using primary hand position-tuned cells. The results shown are based on the same dataset as Fig. 3G. The horizontal scale bar represents 10 s.

Comment 2) The authors should discuss the effects of eye movements especially given that some of the arrays are close to the arcuate sulcus.

Response 2:

We appreciate this helpful suggestion. Accordingly, we have added a part in **Discussion (main text, line 256-264)** to discuss possible effects of eye movements:

In addition to the task-related variables explicitly analyzed in this study, eye movement is a potential confounding factor, as a few electrodes were located near the rostral part of the PMd in our recording, which may engage in oculomotor control². However, we note that majority of electrodes with hand-position tuned cells were located away from the frontal eye fields (see **Supplementary Fig. 9/Response Figure 2** below). In addition, we observed that the monkeys' gazing position during the experiments were primarily around the food reward, rather than their hands, suggesting that the eye movements unlikely have played a significant role in the hand position tuning we observed. Nevertheless, incorporating a gaze fixation protocol or synchronized eye-tracking in future experiments will help further clarify this issue.

Comment 3) Are these hand position tuned neurons spatially organized on their

arrays?

Response 3:

Thanks for the good question. We have added a part in the **Supplementary Information** to assess the spatial organization of the hand position-tuned cells:

To investigate the spatial organization of hand position tuned cells on the arrays, we first set the max firing point in the spatial firing rate map as neuron's hand position preference. Then we labeled the detected hand position preferences for each electrode in the array. To make the pattern more evident, we combined the array results from all four monkeys (**Response Figure 2/Supplementary Fig. 9**).

At each electrode site, if a hand position-tuned cell was recorded and a valid position field was detected, we used colored markers to indicate its positional preference relative to the monkey's body (set as $x=0, y=0$, color white). Red denotes the left-side position of the monkey and green denotes the right-side position of the monkey. Saturation increases from the midline towards the left/right reaching boundaries, and brightness decreases from near area to far area. We analyzed the spatial preferences of all 132-hand position-tuned cells. If multiple valid fields were detected on a single cell, we retained only the field where the cell exhibited maximal firing activities. When merging results from different monkeys, we used the arcuate sulcus (AS) as the reference for alignment.

As illustrated in the figure, red markers dominate at the more dorsal electrodes, while green markers are predominant at the more ventral electrodes. This indicates that, along the dorsal-ventral axis, the preferred hand positions of neurons shift from the left side of the body to the right side. Furthermore, markers near the AS have higher saturation and lower brightness, indicating the position preferences at the reaching limits. Markers away from the AS have lower saturation and higher brightness, indicating that the position preferences locate close to the midline.

Response Figure 2 (Supplementary Fig. 9). Spatial organization of hand position preferences summarized from all monkeys and all sessions. Left panel, circular markers represent hand position preferences of primary hand position-tuned cells ($n =$

50), and square markers represent hand position preferences of remaining hand position-tuned cells ($n = 82$). If multiple hand position-tuned cells were recorded on the same electrode site, the markers sizes are adjusted to ensure that these markers remain within a single grid. Right panel, the marker colors correspond to position preferences relative to the body. Red denotes the left side of the monkey and green denotes the right side of the monkey. Saturation increases from the midline towards the left/right reaching boundaries, and brightness decreases from near area to far area. Monkey's body positions were set as color white.

Comment 4) All in all this is interesting because it uses natural hand tasks to discover position tuning. However, this tuning might be an artifact of the need for a dynamical system to have representations. This might be worth discussing as well as an alternative to their Tolman Eichenbaum machine. As the authors themselves show there ARE a lot of variables being represented. Work from Churchland et al. 2010, and Churchland et al. 2012, Lara et al., Jazayeri lab papers etc, Sussillo et al. 2015, Boucher et al. 2023 etc might be relevant for this contrast.

Response 4:

We appreciate the reviewer's suggestions, and the useful references provided. We added a discussion of the relationship between the position tuning and the neural dynamics in our revision (**main text, line 284-294**):

While our findings emphasize the representational nature of hand position coding in PMd, an alternative interpretation emerges from dynamical systems theories of motor control. Recent studies probing the neural dynamics of PMd and related motor areas³⁻⁷ demonstrate that population activity during reaching reflects low-dimensional, structured manifolds, rather than explicit encoding of kinematic variables. This raises the possibility that the observed hand position tuning may be a consequence of the need for a dynamical system to possess representations. Notably, such dynamics-centric frameworks do not negate the functional relevance of hand position coding. Instead, the mixed selectivity combining hand position tuning with other kinematic tunings may provide a foundation for implementing low-dimensional neural computations⁸. These representations and their geometric structures may serve as initial conditions^{9,10} or boundary constraints that shape the morphology of neural manifolds¹¹.

References

1. Taghizadeh, B., Fortmann, O. & Gail, A. Position- and scale-invariant object-centered spatial localization in monkey frontoparietal cortex dynamically adapts to cognitive demand. *Nat Commun* **15**, 3357 (2024).

2. Fujii, N., Mushiake, H. & Tanji, J. Rostrocaudal Distinction of the Dorsal Premotor Area Based on Oculomotor Involvement. *J Neurophysiol* **83**, 1764 - 1769 (2000).
3. Churchland, M. M. *et al.* Neural population dynamics during reaching. *Nature* **487**, 51 - 56 (2012).
4. Michaels, J. A., Dann, B. & Scherberger, H. Neural Population Dynamics during Reaching Are Better Explained by a Dynamical System than Representational Tuning. *PLoS Comput Biol* **12**, 1 - 22 (2016).
5. Jazayeri, M. & Ostojic, S. Interpreting neural computations by examining intrinsic and embedding dimensionality of neural activity. *Curr Opin Neurobiol* **70**, 113 - 120 (2021).
6. Churchland, M. M. & Shenoy, K. V. Preparatory activity and the expansive null-space. *Nat Rev Neurosci* **25**, 213 - 236 (2024).
7. Denyer, R., Greenhouse, I. & Boyd, L. A. PMd and action preparation: bridging insights between TMS and single neuron research. *Trends Cogn Sci* **27**, 759 - 772 (2023).
8. Fusi, S., Miller, E. K. & Rigotti, M. Why neurons mix: high dimensionality for higher cognition. *Curr Opin Neurobiol* **37**, 66 - 74 (2016).
9. Boucher, P. O. *et al.* Initial conditions combine with sensory evidence to induce decision-related dynamics in premotor cortex. *Nat Commun* **14**, 6510 (2023).
10. Sabatini, D. A. & Kaufman, M. T. Reach-dependent reorientation of rotational dynamics in motor cortex. *Nat Commun* **15**, 7007 (2024).
11. Kriegeskorte, N. & Wei, X.-X. Neural tuning and representational geometry. *Nat Rev Neurosci* **22**, 703 - 718 (2021).